# Reinforcement amid genetic diversity in the *Candida albicans* biofilm regulatory network

Max V. Cravener[1], Eunsoo Do[1], Gemma May[2], Robert Zarnowski[3], David R. Andes[3], C. Joel McManus[2], Aaron P. Mitchell[1]*

1 Department of Microbiology, University of Georgia, Athens, Georgia, United States of America,
2 Department of Biological Sciences, Carnegie Mellon University, Pittsburgh, Pennsylvania, United States of America, 3 Department of Medical Microbiology and Immunology, University of Wisconsin, Madison, Wisconsin, United States of America

* Aaron.Mitchell@uga.edu

**Data Availability Statement:** Processed RNA-Seq data are available in Supporting Information files S3 and S5 Tables; raw data are available through NCBI SRA with accession number PRJNA857655.

## Abstract

Biofilms of the fungal pathogen *Candida albicans* include abundant long filaments called hyphae. These cells express hypha-associated genes, which specify diverse virulence functions including surface adhesins that ensure biofilm integrity. Biofilm formation, virulence, and hypha-associated gene expression all depend upon the transcription factor Efg1. This transcription factor has been characterized extensively in the *C. albicans* type strain SC5314 and derivatives, but only recently has its function been explored in other clinical isolates. Here we define a principal set of Efg1-responsive genes whose expression is significantly altered by an *efg1*Δ/Δ mutation across 17 clinical isolates. This principal gene set includes 68 direct Efg1 targets, whose 5' regions are bound by Efg1 in five clinical isolates, and 42 indirect Efg1 targets, whose 5' regions are not detectably bound by Efg1. Three direct Efg1 target genes encode transcription factors—*BRG1*, *UME6*, and *WOR3*—whose increased expression in an *efg1*Δ/Δ mutant restores expression of multiple indirect and direct principal targets, as well as biofilm formation ability. Although *BRG1* and *UME6* are well known positive regulators of hypha-associated genes and biofilm formation, *WOR3* is best known as an antagonist of Efg1 in the sexual mating pathway. We confirm the positive role of *WOR3* in biofilm formation with the finding that a *wor3*Δ/Δ mutation impairs biofilm formation in vitro and in an in vivo biofilm model. Positive control of Efg1 direct target genes by other Efg1 direct target genes–*BRG1*, *UME6*, and *WOR3*–may buffer principal Efg1-responsive gene expression against the impact of genetic variation in the *C. albicans* species.

## Author summary

Biofilm growth of the fungal pathogen *Candida albicans* is a prevalent source of infection. Biofilm gene regulation is well understood in derivatives of the type strain SC5314, but few studies have examined regulatory features in other clinical isolates. Here we characterize the phenotypic and gene regulatory impact of deletion of master regulatory gene *EFG1* in seventeen clinical isolates that represent a range of clades and infection sites. The

**Funding:** This work was supported by NIH grants 1R01AI146103 (APM) and R01AI073289 (DRA), by the Dr. Frederick A. Schwertz Distinguished Professorship at Carnegie Mellon University (APM), and by startup funds from the University of Georgia (APM). The funders had no role in study design, data collection and analysis, decision to publish, or preparation of the manuscript.

**Competing interests:** The authors have declared that no competing interests exist.

mutation abolishes biofilm formation in all isolates, but has widely variable gene expression impact, even among members of a clade. Fewer than 20% of the gene expression targets in any one isolate are principal targets, which are shared among all isolates. One principal target encodes the transcription factor Wor3, an antagonist of Efg1 in the sexual mating pathway. Our analysis shows that Wor3 has an unexpected role in which it promotes biofilm formation and the expression of principal Efg1-activated genes.

## Introduction

Virtually all microorganisms can grow as surface-associated communities known as biofilms [1]. Biofilm growth offers protection from diverse stresses, including many antimicrobial compounds. Therefore, biofilm growth ability is probably under strong positive selection in the environment. However, both mucosal and deep tissue infections can be outcomes of biofilm formation on tissues and abiotic surfaces [2, 3]. Hence biofilm formation is considered a major challenge in infectious disease and antibiotic development.

Our focus is the fungus *Candida albicans*, a human commensal and pathogen. Biofilm formation by *C. albicans* on medical devices such as venous catheters is a frequent cause of invasive infection [2, 4]. Biofilm formation depends upon production of long filaments called hyphae that grow by tip extension [5]. Hyphal filaments can be hundreds of microns in length. Hyphae contribute to biofilm integrity through the expression of hypha-associated genes, which are expressed at much higher levels in hyphae than in yeast form cells [6]. Several hypha-associated genes specify surface adhesins that contribute to biofilm formation. Hyphae are also required for most forms of infection, and several hypha-associated genes contribute to host tissue damage or virulence [7, 8]. Hyphae and their associated gene expression features are thus integral to *C. albicans* infection biology.

Regulated expression of hypha-associated genes depends upon a large network of transcription factors (TFs) [6, 9]. Among the most well characterized of these TFs is Efg1 [10]. Efg1 binds to the 5' regions of between 446 and 1695 genes, including many hypha-associated genes, in the type strain SC5314 and derivatives [11–13]. Efg1 has additional roles in *C. albicans* biology: it is a negative regulator of the switch from white to mating-competent opaque cells, and a negative regulator of gut colonization and the production of hyper-adapted GUT cells [14]. Some Efg1-bound regions reflect these additional roles. For example, Efg1 binds upstream of *WOR1*, which specifies the central activator of white to opaque switching [14]. Efg1 also binds upstream of *CHT2*, which specifies a chitinase required for gut colonization [12]. These features of the Efg1 regulatory network align well with Efg1 biological functions.

Efg1 is required for hypha and biofilm formation in seven diverse *C. albicans* isolates that have been examined [15–17]. Its biological function shows greater intra-species conservation than other regulators of hypha-associated genes, such as Bcr1 and Ume6, which are dispensable in some strains for hypha and biofilm formation [15]. It seems surprising then that the gene expression changes caused by an *efg1*Δ/Δ null mutation are largely strain dependent. RNA-sequencing (RNA-Seq) comparisons of *efg1*Δ/Δ and *EFG1*+/+ strain pairs derived from five clinical isolates showed that Efg1 affected expression of 523 to 864 genes, depending upon the clinical isolate, and only 177 genes were affected in all five clinical isolates [15]. Additional RNA-Seq replicates brought the number of genes affected in all five clinical isolates up to 200, a small fraction of the total of 1251 genes with Efg1-responsive RNA levels in any of the strains [11]. Chromatin-immunoprecipitation-sequencing (ChIP-Seq) analysis showed that Efg1 is bound to the same genomic sites in all five strains [11], so the differential impact of Efg1 on

gene expression among strains is not a result of differences in its binding sites. Rather, strain differences in expression levels of Efg1 partner TF genes that include *BRG1*, *TEC1*, and *WOR1* are responsible for differential impact of Efg1 on gene expression [11].

The extensive natural variation in Efg1-responsive gene expression raises several fundamental questions. First, is there a principal Efg1 regulatory output that is common to most or all *C. albicans* strains? Does such a principal network include only direct Efg1 target genes, that is, those genes whose 5' regions are bound by Efg1? If the principal network includes indirect Efg1 target genes, what bridges their expression and Efg1 function? And, finally, might the principal network reveal new genes that govern biofilm formation? In this study, we have addressed these questions through analysis of 17 diverse *C. albicans* clinical isolates.

## Results

### Efg1 function in hypha and biofilm formation among clinical isolates

Previous work showed that an *efg1Δ/Δ* mutation blocks hypha and biofilm formation under strong inducing conditions (RPMI+FBS, 37°C) in the five clinical isolates SC5314, P76067, P57055, P87, and P75010 [15]. Here we generated and characterized homozygous *efg1Δ/Δ* mutant strains in 12 additional clinical isolates. All told we utilized 16 clinical isolates sequenced by Hirakawa et al. [16] in addition to the type strain SC5314. This strain set represents five major clades of *Candida albicans* and a range of clinical and geographical origins [18, 19]. Their phenotypic features are diverse as measured by virulence capability in systemic infection models and by strength of virulence-associated traits under in vitro culture conditions [16, 18].

Wild-type and *efg1Δ/Δ* mutant strains in all backgrounds were assayed for planktonic (free-living) hypha formation and for biofilm formation. As observed previously [15, 16], the wild-type strains varied in ability to form planktonic hyphae and biofilm. The *efg1Δ/Δ* mutants were severely defective in formation of planktonic hyphae and biofilm (Figs 1 and 2, S1 and S2). These results indicate that Efg1 is required uniformly for hypha and biofilm formation in all 17 clinical isolates.

### Principal Efg1 target genes among 17 strains

To determine Efg1 regulatory output, RNA-Seq was conducted with all 17 pairs of wild-type and *efg1Δ/Δ* mutant strains under strong hypha-inducing conditions (RPMI+FBS, 37°C). Datasets for the SC5314, P76067, P57055, P87, and P75010 backgrounds have been described previously [11] and are included in the analysis here. Differential expression was calculated for each *efg1Δ/Δ* mutant strain relative to its respective wild type (S1 Table). We found that the *efg1Δ/Δ* mutation altered expression of between 556 and 1270 genes in individual strains, with an average of 777.6 +/- 170.8 genes per strain. The number of Efg1-responsive genes in the type strain SC5314 had suggested that it was an outlier among the strains examined previously [15], but it seems more representative in this larger strain set, having 775 Efg1-responsive genes. Overall, these results point to extensive variation in the number of Efg1-responsive genes among *C. albicans* isolates.

We refer to genes whose expression was regulated by Efg1 in all 17 strains (*efg1Δ/Δ* vs. WT; adjusted $p < 0.05$, fold change $> 1 \log_2$) as principal Efg1 target genes (S1 Table, second tab). The 110 genes in this set included 97 Efg1-activated genes and 13 Efg1-repressed genes. Principal Efg1-activated genes showed significant GO term enrichment for biofilm formation (GOID 44010; corrected P = 1.54E-06) and related terms, and for hyphal cell wall (GOID 30446; corrected P = 2.28E-12) and related terms (S2 Table). These genes include well-studied contributors to hypha or biofilm formation (e.g., *ALS1*, *ALS3*, *HGC1*, *HWP1*, *BRG1*, *UME6*)

## WT

## *efg1Δ/Δ*

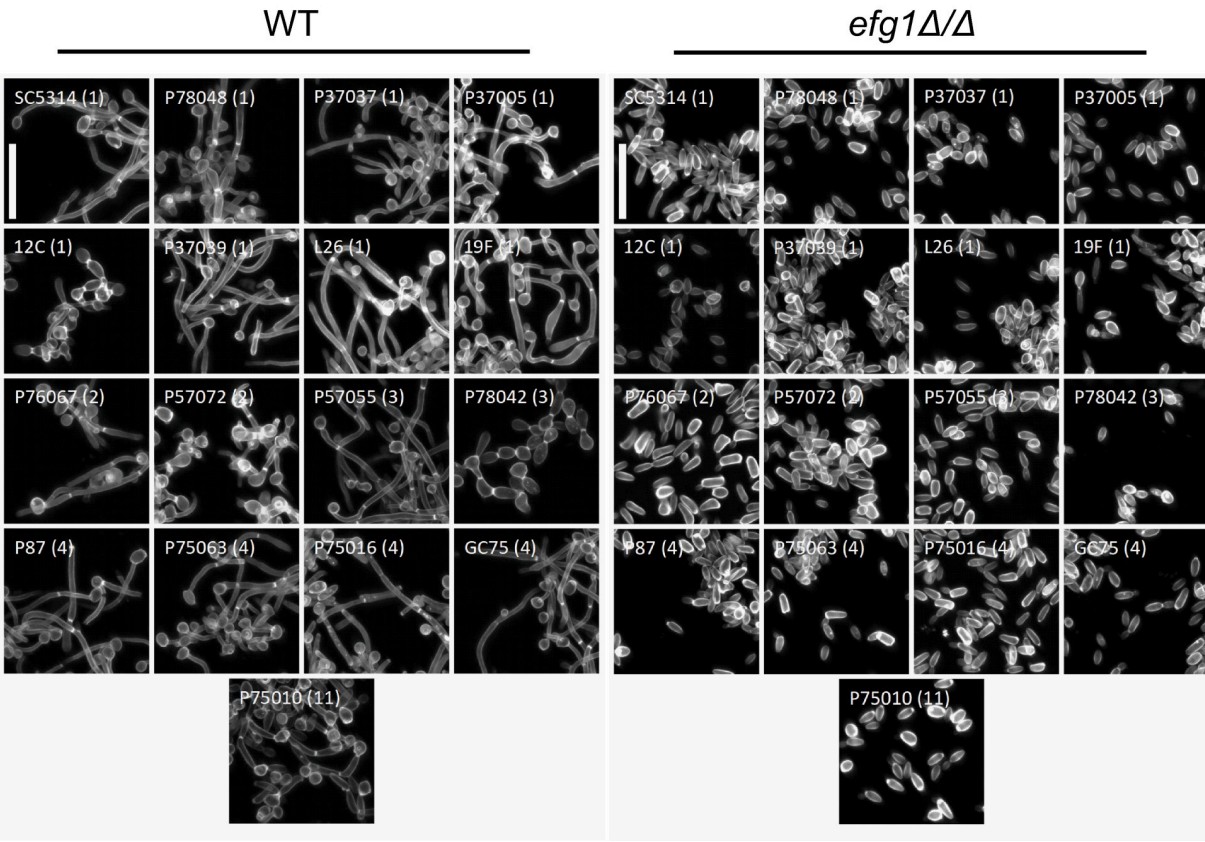

**Fig 1. Hypha formation ability.** Confocal images of clinical isolate WT (Left) and *efg1Δ/Δ* (Right) planktonic hyphal formation in liquid RPMI +10% serum at 37˚C for 4hr. Samples were fixed in 4% formaldehyde in 1X PBS and stained with calcofluor white. White scale bar indicates 25 μm. Clade number is designated in parentheses.

whose 5' regions are bound by Efg1 [11–13]. Principal Efg1-repressed genes showed significant GO term enrichment for cell surface-localized gene products (GOID 9986; corrected P = 0.00041; including *PGA30*, *PGA34*, *RBE1*, *RHD3* and *SCW11*) and related descriptors (S2 Table). Enrichments among principal targets suggest that a central role of Efg1 in these diverse strains is to direct cell surface features.

To explore the relationship between Efg1 targets and hypha-associated genes, we used our profiling and phenotypic data to define a natural variation-based group of hypha-associated genes. We derived a set of 152 genes whose expression levels correlate with hypha formation ability among wild-type strains (S3 Table; see Methods section). This gene set included genes long associated with biofilm and hypha formation (e.g., *ALS3*, *ECE1*, *HWP1*, *UME6*) as well as genes whose connections to these processes are not well understood (e.g., *IHD1*, *IRO1*, *PGA52*, *WOR3*). The 152 hypha-associated genes included 26 principal Efg1 targets (Fig 3). This result suggests that natural variation in hypha-associated gene expression results from variation in outputs of Efg1-dependent and Efg1-independent regulatory pathways.

### Regulation of principal Efg1 indirect target genes

Among the principal Efg1 targets are 68 direct Efg1 targets, which are genes whose 5' regions are bound by Efg1, based on ChIP data [11–13]. The fact that they are regulated

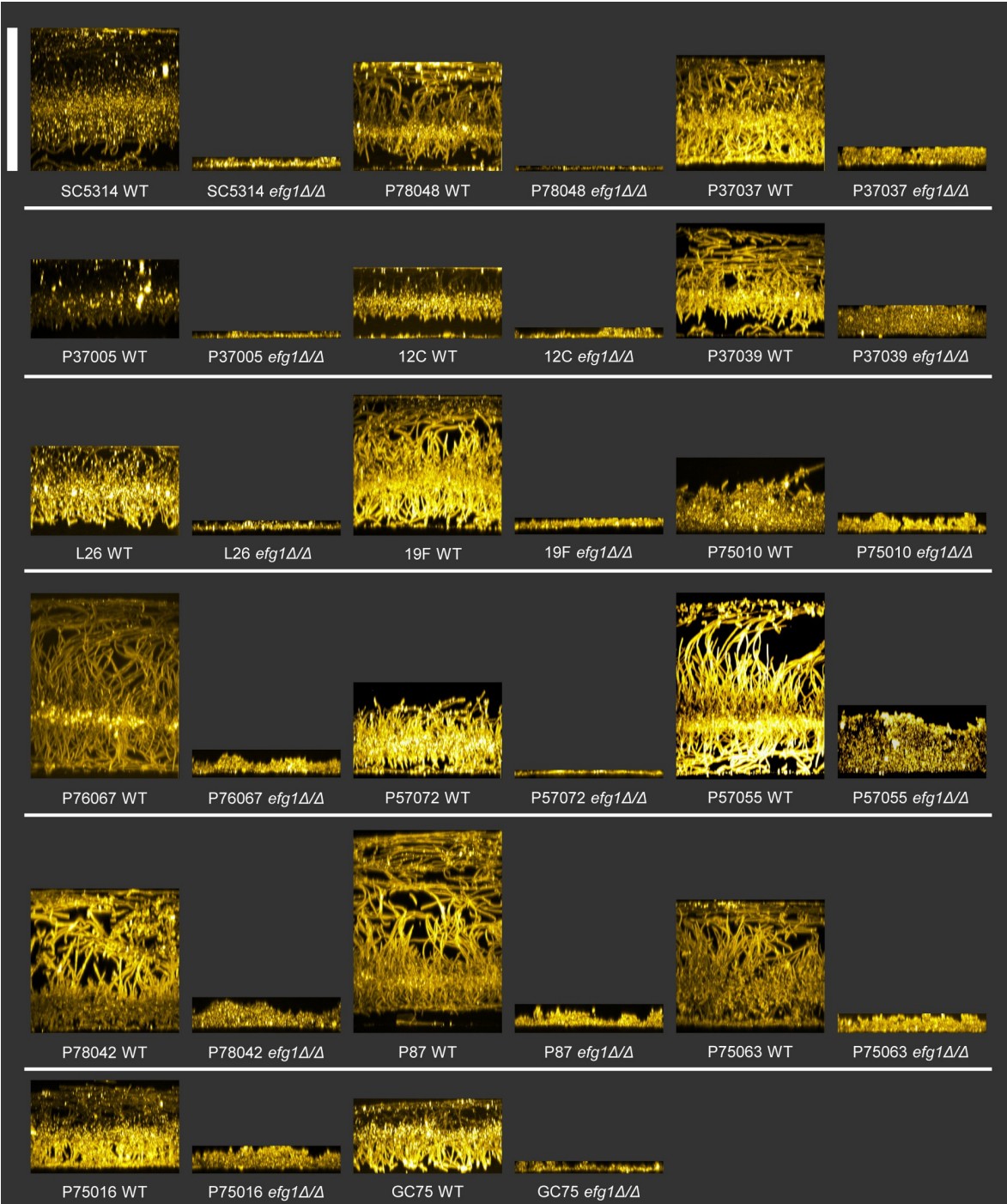

**Fig 2. Biofilm formation ability.** Clinical isolate WT and *efg1Δ/Δ* strains were assayed for biofilm formation ability in RPMI+10% FBS at 37˚C for 24 hrs using the "Silicone substrate–confocal microscopy" method. Specimens were imaged using confocal microscopy and Alexafluor594-conjugated wheat germ agglutinin. Above are sideview images for both WT and *efg1Δ/Δ*. Scale bar indicates 250 μm.

by Efg1 similarly in all 17 strains may reflect uniform features of Efg1 binding, effector activity, and interaction with other TFs across all strains. Another 42 genes are indirect Efg1 targets, which are genes whose 5' regions are not detectably bound by Efg1 [11–13].

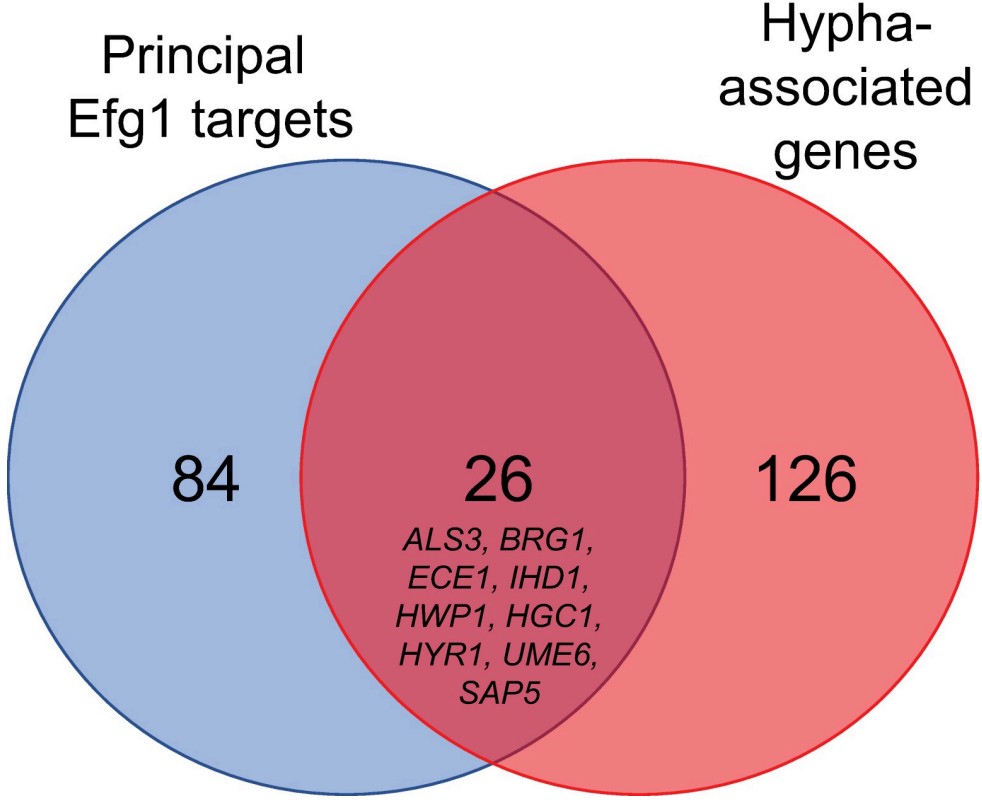

**Fig 3. Relationship between the principal *EFG1* targets and the clinical isolate-derived hypha-associated gene set.**
Well studied hypha-associated genes including *ALS3, BRG1, ECE1, IHD1, HWP1, HGC1, HYR1, UME6,* and *SAP5* are
included in both gene sets.

These genes must be regulated by well-conserved rather than strain-specific network
features.

Among direct principal Efg1-activated targets, four specify TFs: *BRG1*, *ORF19.6888*, *RFX2*
and *UME6*. Three additional direct Efg1-activated targets specify TFs and are uniformly regu-
lated in 16 of 17 strains, but fell slightly below our fold-change cutoff in a 17th strain: *LYS143*,
*TYE7* and *WOR3*. All seven TF genes were downregulated in *efg1Δ/Δ* mutants compared to
respective wild-type strains. If expression of one of these TF genes mediates effects of Efg1 on
some of the Efg1 indirect targets, we predict that overexpression of the relevant TF gene may
reverse effects of an *efg1Δ/Δ* mutation on those Efg1 indirect targets. We increased expression
of each of the seven TF genes through replacement of their 5' regions with *TDH3* promoter
sequences, and introduced the promoter replacements into *efg1Δ/Δ* mutants from three differ-
ent strains–SC5314, P87, and P75010 –in order to identify strain-independent impact of TF
gene overexpression. As a readout, we used Nanostring probes to measure expression of 103
direct and indirect targets under strong hypha-inducing conditions (S4 Table).

*TDH3-BRG1* and *TDH3-UME6* alleles rescued *efg1Δ/Δ* regulatory defects efficiently (Fig 4A
and S4 Table). In the SC5314 *efg1Δ/Δ* mutant background, *TDH3-BRG1* was expressed at
2-fold higher levels than the native *BRG1* allele in wild-type SC5314; *TDH3-UME6* was
expressed at the same levels as the native *UME6* allele in wild-type SC5314 (Fig 4B and S4
Table). *TDH3-BRG1* caused a significant change in RNA levels for 53 of the genes assayed
(excluding *BRG1* itself), including Efg1 direct and indirect targets (Fig 4A and S4 Table).

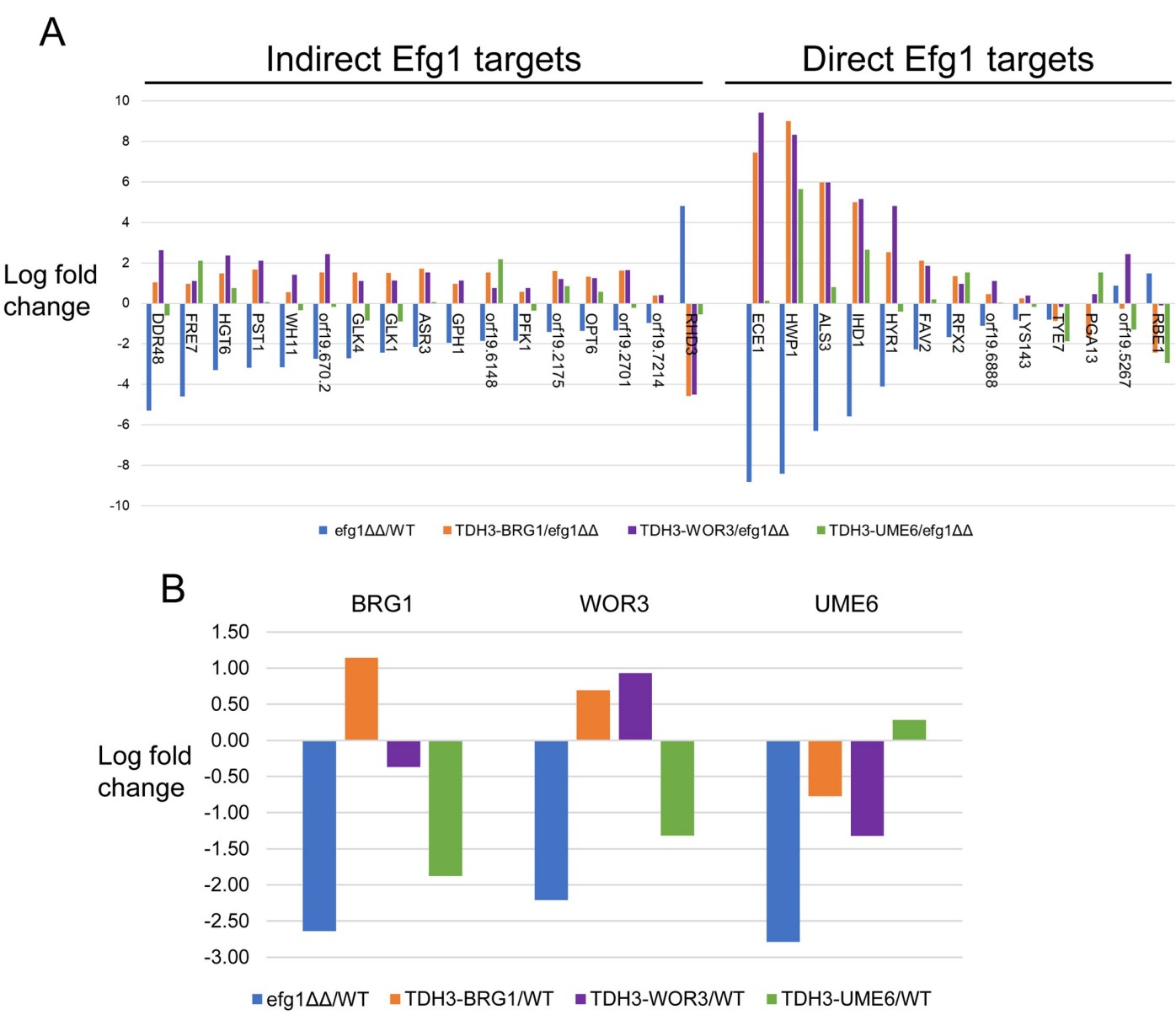

**Fig 4.** Panel A. Relative expression levels of selected 17-strain principal direct and indirect targets. Direct targets are genes whose 5' regions are bound by Efg1; indirect targets are genes whose 5' regions are not detectably bound by Efg1 [11–13]. RNA expression levels were calculated relative to the *efg1*Δ/Δ for each *TDH3-TF* strain under strong hypha-inducing conditions (RPMI+10% FBS at 37˚C). Selected genes were those with a conserved response to a *TDH3-TF* mutation in the SC5314, P87, and P75010 backgrounds. Shown here are the relative expression levels for these genes in SC5314 background. Panel B. Expression levels of *BRG1*, *UME6*, and *WOR3* relative to WT in both the *efg1*Δ/Δ mutant and in each *efg1*Δ/Δ *TDH3-TF* strain in the SC5314 isolate background. P-values for significance are given in S6 Table.

*TDH3-UME6* caused a significant change in RNA levels for 21 of the genes assayed (excluding *UME6* itself), including Efg1 direct and indirect targets (Fig 4A and S4 Table). Effects of *TDH3-BRG1* and *TDH3-UME6* in the SC5314 *efg1*Δ/Δ mutant were supported by single determinations in P87 *efg1*Δ/Δ and P75010 *efg1*Δ/Δ mutants (S4 Table). These results indicate that *BRG1* and *UME6* contribute to expression of both direct and indirect Efg1 targets.

The *TDH3-WOR3* allele also rescued *efg1*Δ/Δ regulatory defects efficiently (Fig 4A and S4 Table). *TDH3-WOR3* was expressed in the SC5314 *efg1*Δ/Δ mutant at 2-fold higher levels than

the native *WOR3* allele in wild-type SC5314 (Fig 4B and S4 Table). *TDH3-WOR3* caused a significant change in RNA levels for 56 of the genes assayed (excluding *WOR3*), including direct targets *ALS3*, *ECE1* and *HWP1*, and indirect targets *DDR48*, *SOD3* and *HGT6* (Fig 4A and S4 Table). Effects of *TDH3-WOR3* were validated with triplicate determinations in P87 *efg1Δ/Δ* and P75010 *efg1Δ/Δ* mutants. Data for significantly affected genes in SC5314 was in good agreement with the other strains (S4 Table). Therefore, *WOR3* contributes to expression of both direct and indirect Efg1 targets in multiple strains.

The *TDH3-LYS143*, *-ORF19.6888*, *-RFX2*, and *-TYE7* alleles had little impact on *efg1Δ/Δ* regulatory defects (S4 Table); the magnitude of most effects was small (S4 Table). Similar effects were observed in the P87 and P75010 strain backgrounds (S4 Table). These results indicate that *LYS143*, *ORF19.6888*, *RFX2*, and *TYE7* contribute narrowly if at all to expression of principal Efg1 targets.

### Restoration of Efg1-dependent phenotypes

Efg1 is required for filamentation and biofilm formation. To determine whether any of the seven TF genes above contribute to Efg1 biological function, we assayed filamentation and biofilm formation in the *efg1Δ/Δ TDH3-TF* strains of all three genetic backgrounds (Figs 5 and S3). *TDH3-UME6* rescued *efg1Δ/Δ* filamentation ability in all three strain backgrounds, in keeping with previous studies in the SC5314 background [20]. *TDH3-BRG1* had a more variable effect; it caused filamentation in SC5314 and P87, but yielded both elongated and misshapen cells in P75010. *TDH3-WOR3* and *TDH3-RFX2* caused slightly elongated cells only in SC5314 (Figs 5 and S3). *TDH3-LYS143* and *TDH3-ORF19.6888* had no effect on filamentation in any strain (S3 Fig). These assays suggest that *UME6*, *BRG1*, *RFX2*, and *WOR3* expression contribute to the connection between Efg1 and cell elongation or filamentation.

Rescue of biofilm formation was surveyed in the SC5314 background. The wild-type strain produced much greater biofilm biomass than the *efg1Δ/Δ* mutant (Fig 6). *TDH3-UME6*, *TDH3-BRG1*, and *TDH3-WOR3* supported biofilm formation comparable to the wild-type strain (Fig 6). The other *TDH3-TF* alleles did not improve *efg1Δ/Δ* biofilm formation. These results argue that *UME6*, *BRG1*, and *WOR3* expression contribute to the connection between Efg1 and biofilm formation.

The structure of *efg1Δ/Δ TDH3-WOR3* biofilms was examined by imaging. In SC5314 the *efg1Δ/Δ TDH3-WOR3* strain produced greater biofilm depth than the *efg1Δ/Δ* mutant and less than the wild type (Fig 7). The *efg1Δ/Δ TDH3-WOR3* biofilm contained mainly yeast-form cells, though some filamentous cells were visible in the upper portion of the biofilm (Fig 7). In P87 the *efg1Δ/Δ TDH3-WOR3* strain also produced greater biofilm depth than the *efg1Δ/Δ* mutant, and some upper-layer filamentous cell content was evident (S4 Fig). In P75010 the *efg1Δ/Δ TDH3-WOR3* strain yielded greater biofilm cell density than the *efg1Δ/Δ* mutant; filamentation levels were low, as they were in the wild type (S4 Fig). These results indicate that *WOR3* expression can promote both adherence and filamentation in the absence of Efg1, an observation that strengthens the functional parallels between *WOR3* and *EFG1*.

### Positive role of *WOR3* in biofilm formation

*WOR3* has well known roles in white-opaque switching and gut commensalism [12, 14], but has not been implicated previously in biofilm formation. Gene expression and phenotypic assays of *TDH3-WOR3* strains above argue that *WOR3* may be a positive regulator of Efg1-dependent genes and biofilm formation. To test this role of *WOR3*, we asked whether loss of *WOR3* function affects biofilm formation. We constructed homozygous *wor3Δ/Δ* mutant strains in the SC5314 and P87 backgrounds, and created complemented strains with the

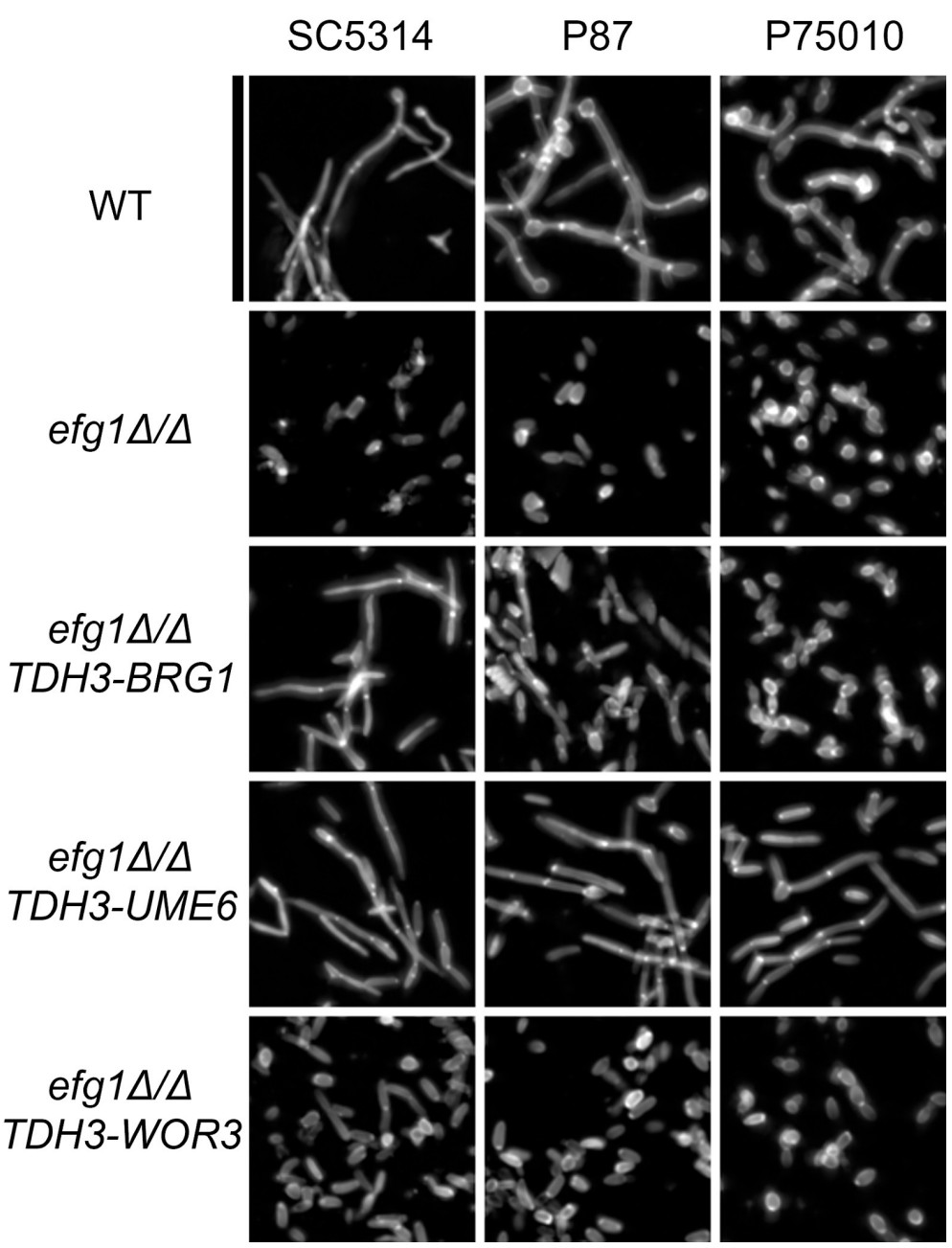

**Fig 5. Rescue of *efg1Δ/Δ* hypha formation.** Fluorescence image of SC5314, P87, and P75010 wild-type (WT), *efg1Δ/Δ*, *efg1Δ/Δ TDH3-BRG1*, *efg1Δ/Δ TDH3-UME6*, and *efg1Δ/Δ TDH3-WOR3* strain planktonic hypha formation in RPMI +10%FBS 37˚C for 4hrs. Samples were fixed in 4% formaldehyde in 1X PBS and stained with Calcofluor-white. Scale bar indicates 80 μm.

*TDH3-WOR3* allele. The *wor3Δ/Δ* mutation caused a prominent defect in biofilm formation in RPMI+FBS in both strain backgrounds (Fig 8A and 8B). The defect was reversed by the *TDH3-WOR3* allele (Fig 8A and 8B). The *wor3Δ/Δ* mutants had a less pronounced defect under weaker inducing conditions, including RPMI without FBS as well as YPD+10% FBS (Fig 8A and 8B). Our results indicate that *WOR3* has a positive role in biofilm formation in vitro, and that its impact depends upon growth conditions.

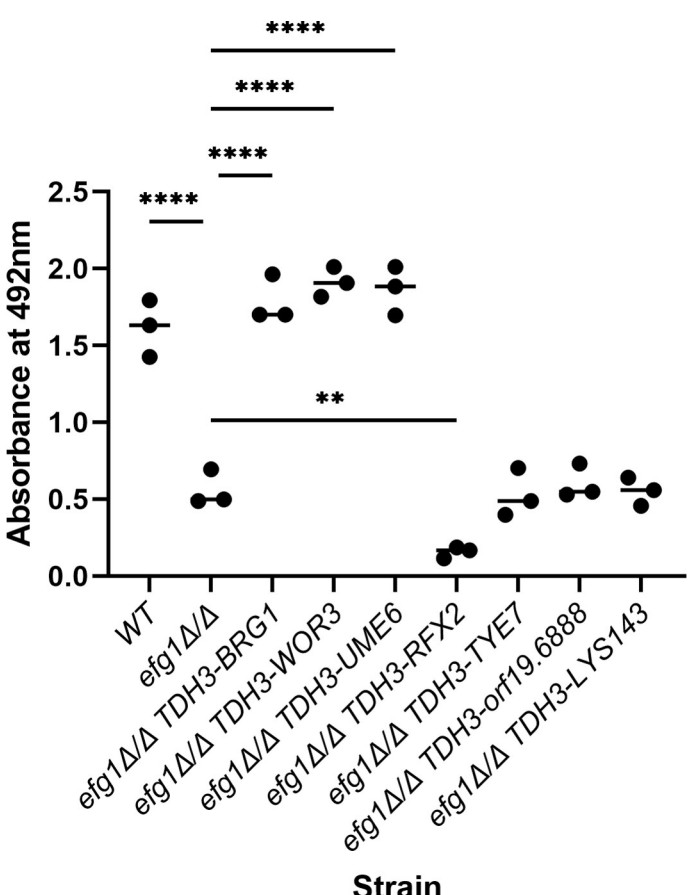

**Fig 6. Rescue of *efg1Δ/Δ* biofilm formation.** Restoration of biofilm formation by Efg1 target TF gene expression. Biofilm formation was assayed in RPMI+10% FBS at 37°C for 24 hrs in a 96 well plate for WT, *efg1Δ/Δ*, and *efg1Δ/Δ TDH3-TF* strains in the SC5314 background using the "96 well plate—XTT reduction" method. After growth, wells were washed with PBS and 100μL of 1mg/mL XTT in PBS plus 0.025 μL menadione in acetone were added to each well and incubated for 1 hour at 37°C. Absorbance at 492nm was then measured to quantify biofilm formation. Three technical replicates for each strain were compared by one-way ANOVA to the *efg1Δ/Δ* strain. ** denotes p = 0.0080 and **** denotes p < 0.0001.

Given that the *wor3Δ/Δ* biofilm defect phenotype depends upon growth conditions, we sought to determine whether *WOR3* may be required in an animal biofilm infection model. We used SC5314-derived wild-type, mutant, and complemented strains in a rat venous catheter infection model [21]. *C. albicans* cell counts at 48 hr post-infection indicated that the *wor3Δ/Δ* mutation causes a reduction in biofilm population size (Fig 8C). These results indicate that *WOR3* has a positive role in biofilm formation in vivo, as it does in vitro.

## Relationship of *WOR3* and *BRG1* in biofilm formation

We considered the possibility that Wor3 promotes biofilm formation solely through activation of *BRG1* expression. This hypothesis was based upon the observations above that gene expression impact of *TDH3-WOR3* and *TDH3-BRG1* in *efg1Δ/Δ* strains is similar, and that *BRG1* RNA levels are up-regulated in *efg1Δ/Δ TDH3-WOR3* strains (Fig 4B and S4 Table). Also, published ChIP-Chip data show that Wor3 is associated with the *BRG1* 5' region [22, 23]. This hypothesis predicts that Wor3 will be incapable of stimulating biofilm formation in the absence of functional Brg1. We tested this prediction with *brg1Δ/Δ* mutants of strains SC5314,

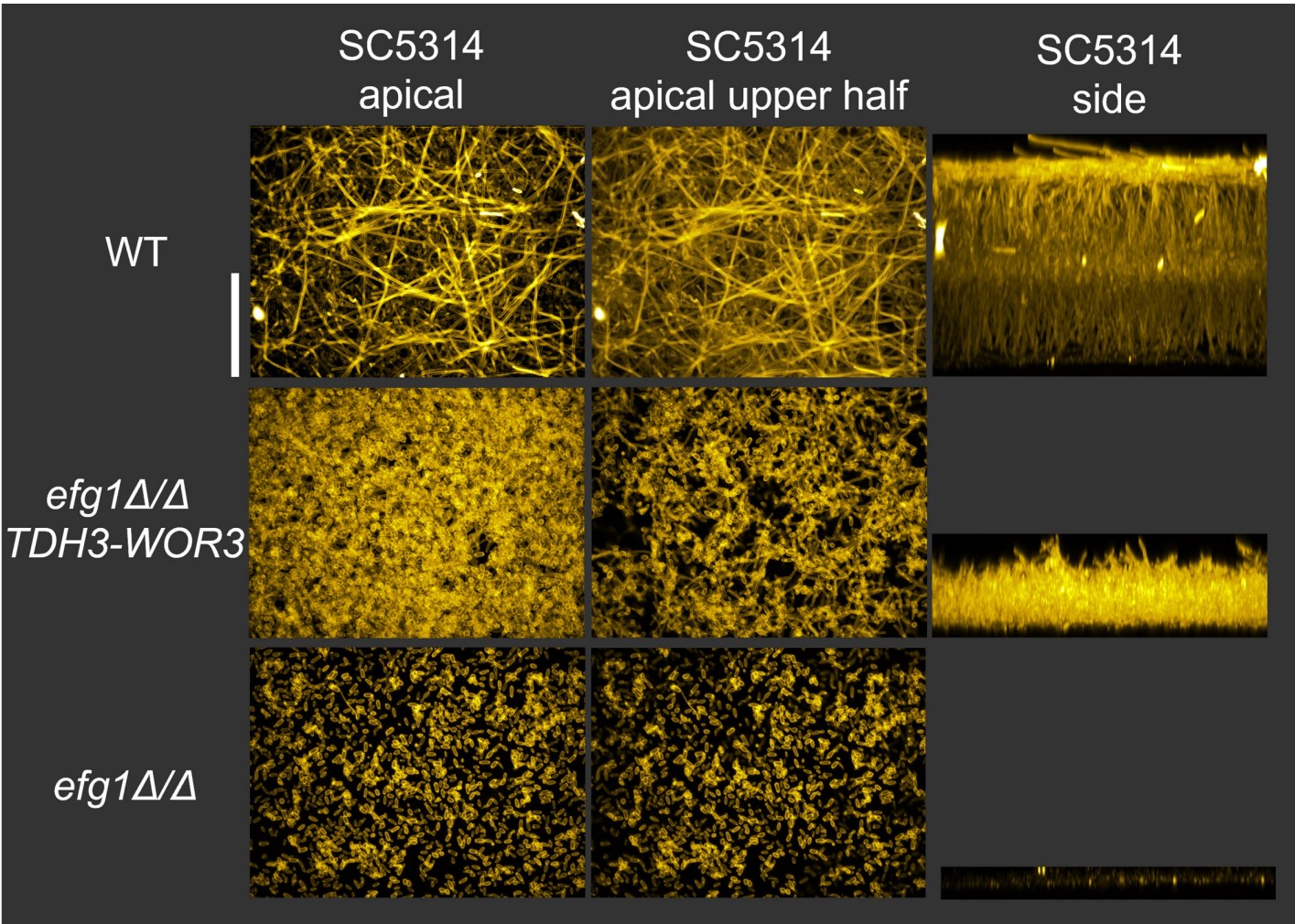

**Fig 7. Imaging of *TDH3-WOR3* biofilms in the SC5314 background.** Biofilms were imaged after 24 hrs at 37˚C in RPMI+10% FBS for WT, *efg1Δ/Δ*, and *efg1Δ/Δ TDH3-WOR3* SC5314-derived strains using the "Silicone substrate–confocal microscopy" method. Maximum intensity apical projections of the entire biofilm (Left column) and the top half of the biofilm (Middle column) were generated for each strain. Side view projections (Right column) were also generated. Scale bar indicates 60 μm.

P87, and P75010. The *brg1Δ/Δ* mutants were biofilm-defective compared to their parent strains; introduction of the *TDH3-WOR3* allele improved biofilm formation dramatically (Fig 9). We conclude that Wor3 can promote biofilm formation independently of *BRG1*.

## Discussion

Biofilm and hypha formation are central pathogenicity traits of *C. albicans*. Efg1 is among the most extensively characterized regulators of these processes [10], yet it has been studied almost exclusively in the type strain SC5314 and derivatives. Our previous studies of Efg1 in five clinical isolates showed that its gene expression output varies considerably among strains [15]. Here we have extended our phenotypic and gene expression analysis to 17 clinical isolates, including multiple representatives of several clades. Our findings indicate that Efg1 is required uniformly for biofilm and hypha formation, but that its gene expression output is highly variable. In fact, the principal Efg1-responsive genes, which are common to all strains tested,

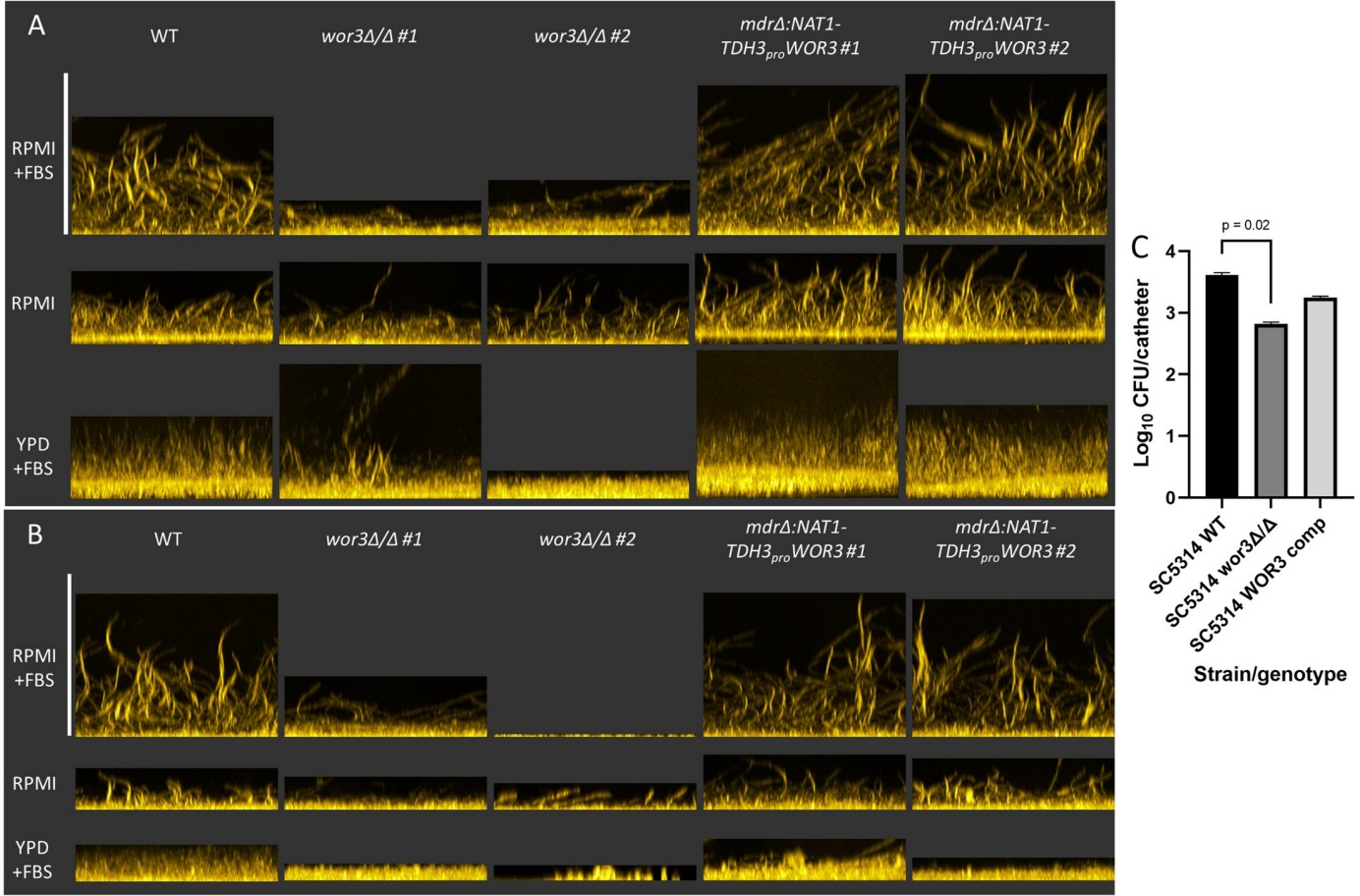

**Fig 8. Biofilm formation dependence on *WOR3*.** Panels A and B. Biofilm formation ability for WT, *wor3Δ/Δ*, and complemented strains was assessed at 37°C for 24 hrs in RPMI, RPMI+10% FBS, and YPD+10% FBS in both SC5314 (Panel A) and P87 (Panel B) backgrounds using the "96 well plate–Fluorescence microscopy" method. Side-view images of two independent mutant strains were tested for each clinical isolate background for both the *wor3Δ/Δ* and complemented strains. Scale bars are equal to 285 μm. Panel C. SC5314-derived wild-type, mutant, and complemented strains were tested for biofilm formation in a rat venous catheter infection model [21]. *C. albicans* cell counts per catheter were determined at 48 hr post-infection. Wild-type and mutant strains differed with p-value = 0.02.

comprise fewer than 20% of the Efg1-responsive genes in any one strain. Our analysis of the principal genes led to the identification of a new biofilm regulator, Wor3. Wor3 is best known as a positive regulator of white-opaque switching, a process that is negatively regulated by Efg1 [14]. Our findings here show that Wor3 "switches sides" to act in conjunction with Efg1 in support of biofilm formation and activation of principal Efg1 target genes. Wor3 is also known to act in conjunction with Efg1 as a negative regulator of gut colonization [12], which also may reflect their interwoven network relationship. Our findings echo the recent discovery that another white-opaque activator, Wor1, also acts in conjunction with Efg1 to promote biofilm formation [11].

## Natural variation in Efg1 outputs

The *efg1Δ/Δ* mutant biological phenotype is consistent among strains: there is a severe defect in biofilm and hypha formation. In contrast, the *efg1Δ/Δ* mutant gene expression phenotype is highly variable: an average of 777.6 genes responded significantly to an *efg1Δ/Δ* defect in any one strain, yet only 110 genes responded to an *efg1Δ/Δ* defect in all 17 strains. Genes that did

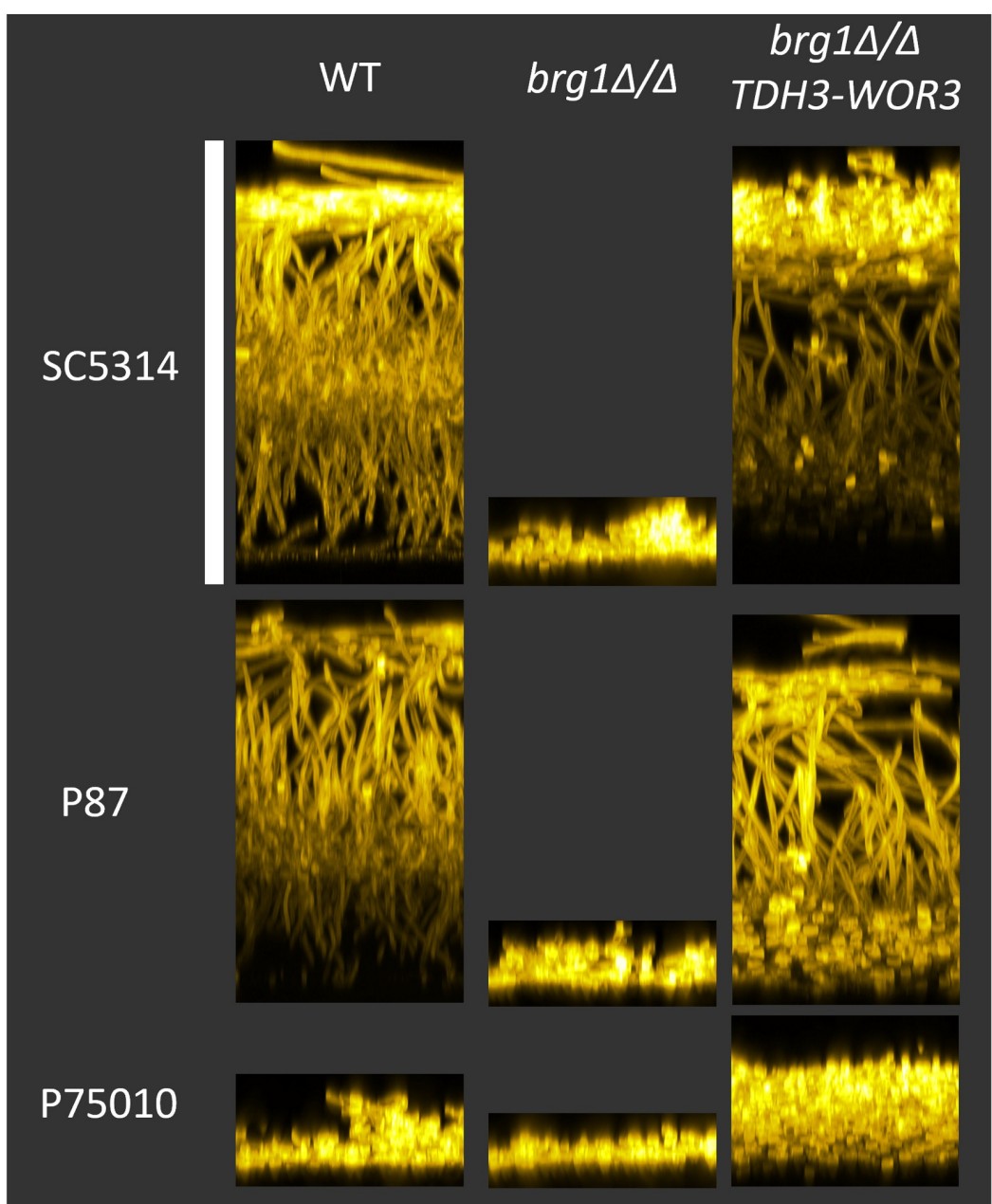

**Fig 9. Biofilm formation in *brg1Δ/Δ* and *brg1Δ/Δ TDH3-WOR3* strains.** Wild-type (WT), *brg1Δ/Δ*, and *brg1Δ/Δ TDH3-WOR3* strains in the indicated clinical isolate backgrounds were assayed for biofilm formation in RPMI+FBS, 37˚C for 24hrs using the "Silicone substrate–confocal microscopy" method. Biofilm sideview images are above. Scale bar indicates 145 μm.

not respond consistently to *efg1Δ/Δ* defects include many that respond strongly in the type strain SC5314. The consistently responsive genes, or principal Efg1 targets, are highly enriched for genes with roles in biofilm formation and aggregation, and thus are tightly connected to the uniform *efg1Δ/Δ* mutant phenotypes.

How does such extensive strain variation among Efg1-responsive genes arise? Our recent studies indicate that small expression differences for any of three "partner" TF genes–*BRG1*,

*TEC1*, or *WOR1* –can determine which genes respond to Efg1 and how they respond [11]. For example, increased expression of *WOR1* in strain P75010 places expression of *CHT2* under negative control rather than positive control by Efg1 [11]. Overlap in the gene expression impact of *BRG1*, *TEC1*, and *WOR1* suggests that they may augment each other's effects. The new datasets we describe here yield a wealth of additional TF genes whose strain-to-strain expression variation may translate in strain-specific differences in regulatory networks. For example, 6–8 fold differences in RNA levels for *RME1* and *CRZ2* may account for some of the Efg1 network differences between strains SC5314 and P57072 or P76067; 3–4 fold differences in RNA levels for *NRG1* may account for some of the Efg1 network differences between strains SC5314 and P75016 or P78042 (S3 Table). Although our focus here was on the commonalities among strains, this discussion illustrates that our data contribute candidate genes for future dissection of strain-to-strain network variation.

Efg1 network variation may contribute to phenotypic variation. The natural variation-based group of hypha-associated genes includes several Efg1 targets that contribute to biofilm formation, such as *ALS3* and *HWP1* [2]. It also includes Efg1 targets *BRG1* and *WOR3*, whose products activate *ALS3* and *HWP1* expression ([13] and this work). *BRG1* and *WOR3* expression respond to many different signals [6, 14], so it is likely that combined outputs of the Efg1 network and of Efg1-independent networks together contribute to phenotypic variation.

## Utility of principal target definition

For many regulatory genes, differential expression comparison of wild-type and deletion mutant strains in a single strain background can yield many regulated target genes. Combining expression and chromatin binding data narrows the number of target genes, but still many targets may be irrelevant to the biological process under study. Based on our results here, it may help define functionally relevant targets of any TF to compare wild-type and mutant expression profiles in multiple strains. How many strains should one examine to optimize effort? We developed a regression model of the size of the principal gene set as a function of the number of strains examined to help address this question (Fig 10A). Comparison of wild-type and *efg1*Δ/Δ strains in any one background is expected to yield ~778 Efg1-responsive genes. Extending the comparison to a second background reduces that number by 349 genes, or 45%. Extending the comparison to a third background reduces the number by 106 genes, or 30% of the number of 2-strain principal targets. Hence inclusion of only a few strains has potential to narrow the focus to functionally relevant genes substantially. Extending the comparison beyond six backgrounds yields less than a 10% improvement for each additional strain. Similar outcomes occur if the gene expression cutoff is increased to a 4-fold or 8-fold change (Fig 10B). Although the detailed cost-benefit balance may be specific to any particular study, our results with Efg1 indicate that analysis of between three and six backgrounds is worthwhile to narrow the set of principal targets; beyond six backgrounds there are diminishing returns.

## Architecture of the 17-strain principal Efg1 regulatory network

Expression of the 110 principal Efg1 target genes is defined by expression alterations in the same direction in *efg1*Δ/Δ mutants of all strains. Among those consistently Efg1-responsive genes, 42 genes are indirect targets, in that their 5' regions are not detectably bound by Efg1. One simple explanation for the consistent responses of these genes is that they are targets of a TF whose expression is directly regulated by Efg1; the TF is an intermediary that relays the Efg1-dependent signal. Restoration of approximately wild-type expression levels of individual candidate TF genes in an *efg1*Δ/Δ background led to two major conclusions (Fig 11). First, Brg1 and Wor3 are the major intermediaries that link Efg1 activity to expression of Efg1

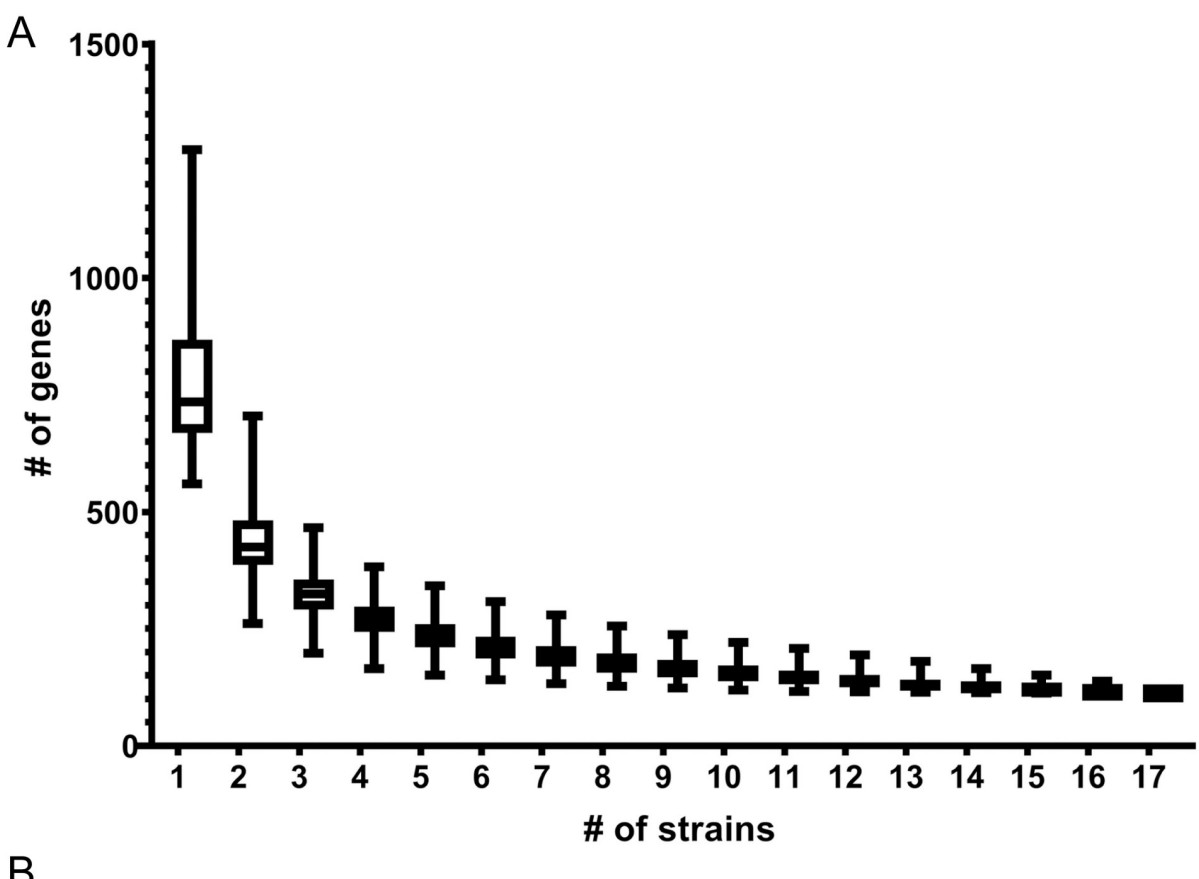

**Fig 10. Size of *EFG1* responsive gene set.** Panel A. Regression showing the decay of the number of principal *EFG1*-responsive genes with the addition of multiple strains. Each box plot represents all possible number of principal response genes (Y-axis) among a given number "n" of strains (X-axis). Panel B. Tabulation of mean number of Efg1-responsive genes as a function of number of strains tested with fold changes of 1 $\log_2$ (as used throughout this study; described by $y = 701.93x^{-0.656}$ and $R^2 = 0.9894$), or with higher thresholds of 2 $\log_2$ (described by $y = 310.83x^{-0.717}$ and $R^2 = 0.9974$) or 3 $\log_2$ (described by $y = 180.96x^{-0.844}$ and $R^2 = 0.999$).

| Cut-off | Number of Strains: | 1 | 2 | 3 | 4 | 5 | 6 | 7 | 8 | 9 | 10 | 11 | 12 | 13 | 14 | 15 | 16 | 17 |
|---|---|---|---|---|---|---|---|---|---|---|---|---|---|---|---|---|---|---|
| | Mean | 777.6 | 428.8 | 322.2 | 267.7 | 233.2 | 208.7 | 190.2 | 175.5 | 163.4 | 153.3 | 144.6 | 137.1 | 130.4 | 124.5 | 119.2 | 114.4 | 110 |
| 1log₂ | Std. Deviation | 170.8 | 64.9 | 46.76 | 39.1 | 34.24 | 30.61 | 27.62 | 25 | 22.61 | 20.36 | 18.17 | 16.01 | 13.83 | 11.57 | 9.139 | 6.393 | 0 |
| | Std. Error of Mean | 41.43 | 5.565 | 1.793 | 0.8014 | 0.4353 | 0.2751 | 0.198 | 0.1604 | 0.145 | 0.146 | 0.1633 | 0.2035 | 0.2835 | 0.4437 | 0.7837 | 1.55 | 0 |
| | Mean | 325.1 | 181.8 | 135.5 | 110.8 | 94.93 | 83.6 | 74.98 | 68.14 | 62.56 | 57.91 | 53.97 | 50.58 | 47.63 | 45.05 | 42.77 | 40.76 | 39 |
| 2log₂ | Std. Deviation | 71.71 | 33.71 | 26.41 | 22.78 | 20.51 | 18.81 | 17.36 | 16.01 | 14.68 | 13.35 | 11.99 | 10.6 | 9.153 | 7.629 | 5.986 | 4.146 | 0 |
| | Std. Error of Mean | 17.39 | 2.891 | 1.013 | 0.467 | 0.2607 | 0.1691 | 0.1245 | 0.1027 | 0.0942 | 0.0957 | 0.1078 | 0.1347 | 0.1876 | 0.2926 | 0.5133 | 1.006 | 0 |
| | Mean | 182.5 | 95.22 | 67.82 | 53.67 | 44.71 | 38.39 | 33.63 | 29.89 | 26.85 | 24.34 | 22.23 | 20.41 | 18.84 | 17.45 | 16.2 | 15.06 | 14 |
| 3log₂ | Std. Deviation | 44.09 | 19.17 | 14.79 | 12.4 | 10.72 | 9.399 | 8.312 | 7.387 | 6.583 | 5.869 | 5.223 | 4.622 | 4.042 | 3.454 | 2.812 | 2.045 | 0 |
| | Std. Error of Mean | 10.69 | 1.643 | 0.5671 | 0.2543 | 0.1362 | 0.0845 | 0.0596 | 0.0474 | 0.0422 | 0.0421 | 0.047 | 0.0588 | 0.0829 | 0.1325 | 0.2411 | 0.4961 | 0 |

indirect targets under our growth conditions. Each TF governs expression of between 20% and 75% of principal Efg1 indirect targets, depending on the strain background. Second, Brg1, Wor3, and to a lesser extent Ume6 control both indirect and direct targets of Efg1. Thus Brg1, Wor3, and Ume6 act both to extend and to reinforce the Efg1 network through activation of both indirect and direct Efg1 target genes.

What is the significance of "reinforcement?" The connections among Efg1, Brg1, Wor3, Ume6, and principal direct Efg1 targets create three feed-forward loops [24]: Efg1-Brg1-principal direct targets, Efg1-Wor3-principal direct targets, and Efg1-Ume6-principal direct targets. It is well appreciated that feed-forward loops exist in the *C. albicans* biofilm regulatory

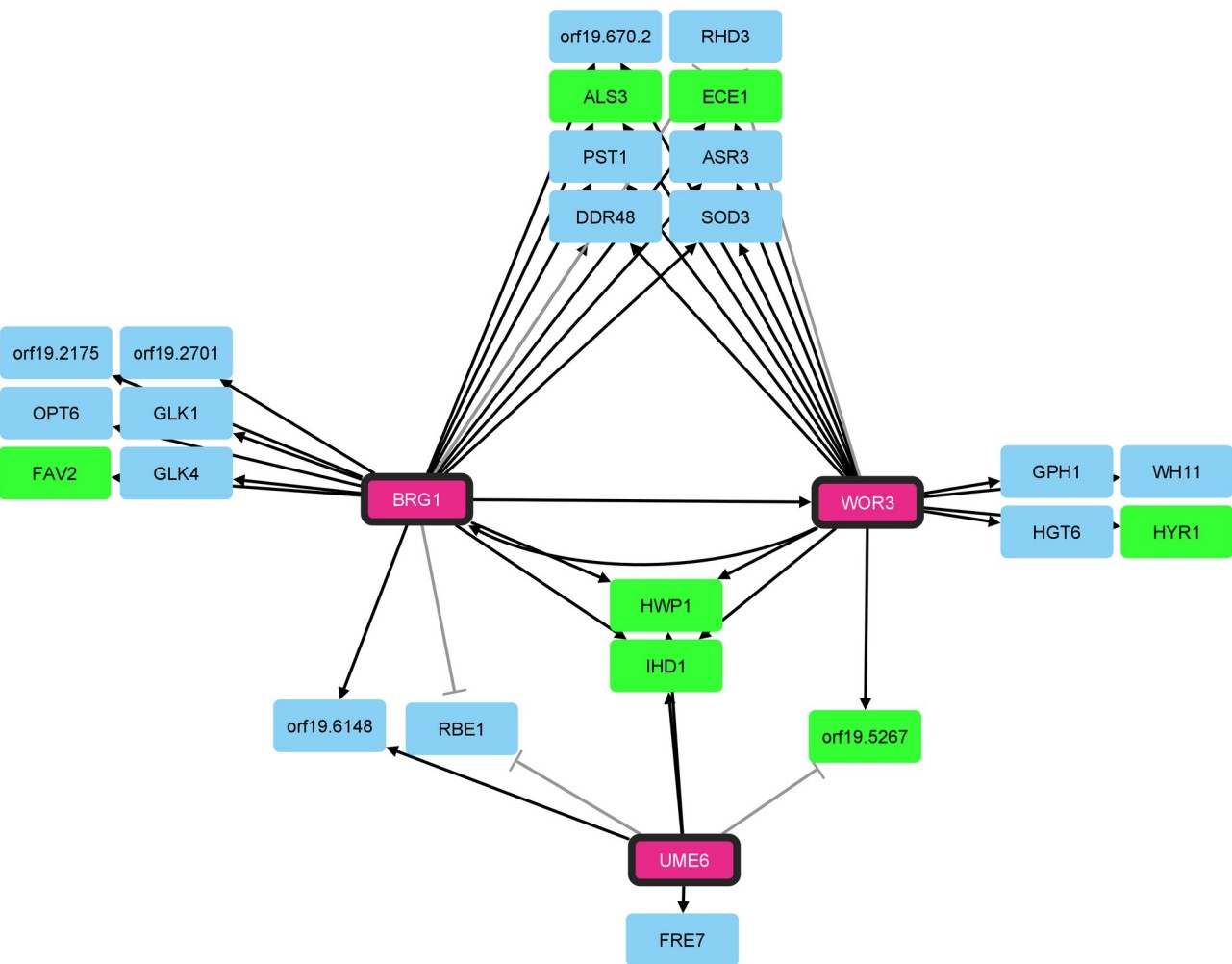

**Fig 11. Reinforcement and extension of the Efg1 principal network by *BRG1*, *UME6*, and *WOR3*.** Network of Efg1 direct target transcription factor interactions with other principal Efg1 direct targets (green) and Efg1 indirect targets (blue). Arrows represent conserved activation by the *TDH3-TF* allele in the SC5314, P87, and P75010 *efg1Δ/Δ* strain backgrounds. Blunt-end connections represent repression by the *TDH3-TF* allele in all three clinical isolate *efg1Δ/Δ* backgrounds. Bolded arrows represent conserved activation among all three clinical isolate *efg1Δ/Δ* backgrounds of a principal Efg1 direct target TF by one of the other principal Efg1 direct target TFs. Network relationships were deduced from one set of growth conditions and three diverse *C. albicans* strain backgrounds.

network [25, 26], and more generally that feed-forward loops can alter kinetics of gene expression responses [24, 25]. Here our data argue for something different: that the ability of Brg1, Wor3, and Ume6 to reinforce expression of direct Efg1 target genes maintains stability of the principal Efg1 network in the face of natural genetic variation. This inference comes from the demonstration that the Efg1-principal target relationship is a constant among 17 diverse strains, and that reconstituted expression of *BRG1*, *WOR3*, or *UME6* in *efg1Δ/Δ* mutants of three diverse backgrounds restores expression of several principal Efg1 targets. It seems unlikely that any strain lineage would be subject to a sequence of forward and back DNA sequence changes that would impose selections for reinforcement. However, a series of

chromosome gain and loss events could occur in a lineage in response to naturally encountered stresses [27]. Reinforcement of the Efg1 network may have been selected to safeguard critical gene expression output from disruptive effects of intermittent aneuploidy.

## Wor3 function in biofilm formation

Our analysis led to a new positive regulator of biofilm formation, Wor3. The evidence in support of this function is that a *wor3Δ/Δ* mutation causes reduced biofilm formation, that the *wor3Δ/Δ* biofilm defect is reversed when *WOR3* expression is restored, and that these genetic benchmarks have been met in two different genetic backgrounds. In addition, expression of *WOR3* in an *efg1Δ/Δ* mutant is sufficient to promote biofilm formation. However, *WOR3* is not required for biofilm formation under all growth conditions. *WOR3* was not included in some earlier screens for biofilm [13] or adherence [28] mutants, and may have been undetected in others [29] due to the medium used for assays. It has become increasingly evident that the strength of several biofilm and hyphal mutant phenotypes can vary extraordinarily with growth conditions [15, 30, 31]. Perhaps functional redundancy of Wor3 with some other TF is the basis for the *wor3Δ/Δ* mutants' environmentally contingent phenotype. Overlap among gene expression targets (Fig 11) suggests that Brg1 and Wor3 may be functionally redundant under some growth conditions.

How does Wor3 promote biofilm formation? Biofilm production in an *efg1Δ/Δ* background requires increased adherence [13, 32], likely a consequence of increased *ALS3* and *HWP1* expression in response to *TDH3-WOR3* (Fig 11). Neither *ALS3* nor *HWP1* is a direct Wor3 target, based on ChIP analysis in opaque cells [22, 23], so Wor3 may act indirectly through control of other TF genes. We infer that Wor3 does not act solely through its direct target *BRG1* because *TDH3-WOR3* drives biofilm formation in *brg1Δ/Δ* mutants. However, Wor3 also binds to the *UME6* 5' region [22, 23], so Wor3 may act through both *BRG1* and *UME6*.

What signal may determine the functional relationship between Wor3 and Efg1? Wor3 opposes Efg1 in driving the switch from white to opaque [14]. Wor3 acts in conjunction with Efg1 to promote biofilm formation. Wor3 and Efg1 also both promote GI tract colonization [12], though it is not yet known whether they act in the same gene regulatory pathway in that context. White-opaque switching occurs most frequently at temperatures below 30˚C, whereas biofilm formation and GI tract colonization occur at 37˚C [14]. Two key players in the heat shock response, chaperone Hsp90 and transcription factor Hsf1, also govern expression of biofilm- and hypha-related genes [33]. A simple model is that Hsp90, Hsf1, or their downstream targets interact with Wor3 to alter its activity.

## Concluding remarks

Our immediate goals in this and other recent studies [11, 15] have been to use strain variation to understand gene function, and to use gene function to understand strain variation. The first goal has been realized here through the finding that Wor3 functions in the Efg1 regulatory network to promote biofilm formation. The second goal has been realized here through the finding that there is a principal Efg1 network that is quite small, comprising at most 20% of the Efg1-responsive genes in any one strain, and only 60% of the Efg1-responsive genes common to five strains examined previously. Does our study change the broader understanding of the *C. albicans* biofilm regulatory network [9]? Our results emphasize the issue that biofilm formation and its control are condition-dependent [34]. This point has been acknowledged previously in the context of environmental variation [6, 9], but to our knowledge there was limited appreciation of the overarching impact of *C. albicans* strain variation on biofilm circuitry prior to the present study and our two related reports [11, 15]. Definition of the constant features of

biofilm formation among diverse *C. albicans* isolates offers a simple rationale to prioritize circuits and targets for in depth analysis. Finally, we note that similar approaches to ours could be applied to gain better understanding of any regulatory network, and to discover new regulators that direct network behavior, including features that differ among strains and features common to all strains.

## Materials and methods

### Strains and culture conditions

Clinical isolates [16] were obtained from BEI resources NIAID, NIH: *Candida albicans*, Strain GC75, NR-29452; *Candida albicans*, Strain P37039, NR-29451; *Candida albicans*, Strain 12C, NR-29444; *Candida albicans*, Strain P37005, NR-29447; *Candida albicans*, Strain P37037, NR-29450; *Candida albicans*, Strain P75016, NR-29438; *Candida albicans*, Strain L26, NR-29445; *Candida albicans*, Strain P57072, NR-29345; *Candida albicans*, Strain 19F, NR-29449; *Candida albicans*, Strain P75063, NR-29440; *Candida albicans*, Strain P75010, NR-29437; *Candida albicans*, Strain P57055, NR-29439; *Candida albicans*, Strain P76067, NR-29442; *Candida albicans*, Strain P87, NR-29453; *Candida albicans*, Strain P78048, NR-29434; *Candida albicans*, Strain P78042, NR-29443. Long term storage of strains was in 15% glycerol solution at -80°C. Strains were grown prior to all experiments on YPD (2% peptone, 2% dextrose, 1% yeast extract) solid medium (2% agar) at 30°C for 48 hrs and cultured overnight in YPD liquid medium at 30°C with agitation. Transformant colonies were selected on YPD + 400μg/mL nourseothricin or CSM-His (1.7% Difco yeast nitrogen base with ammonium sulfate with amino acid supplement lacking histidine, 2% dextrose, 2% agar). Planktonic filamentation, biofilm formation, and RNA cell cultures were conducted in liquid RPMI-1640 media (Sigma-Aldrich, Inc., St. Louis) with pH 7.4 with 10% fetal bovine serum (Atlanta Biologicals, In., Flower Branch). All strains used in this work and their genotypes can be found in the S5 Table.

### Transformation of *C. albicans* strains

Transformations were done in accordance with the transient CRISPR protocol [35] in strains P78048, P37037, P37005, 12C, P37039, L26, 19F, P57072, P34048, P75063, P75016, and GC75. In brief WT strains were made auxotrophic for histidine via homozygous deletion of the *HIS1* coding sequence with a recyclable nourseothricin resistance marker, (*NAT1* amplified from pMH05 and pMH06) with 80-300bp of flanking homology to the *HIS1* up and downstream regions [15, 36]. These *his1Δ::r3NAT1r3* strains were then transformed to generate a homozygous deletion of the *EFG1* coding sequence with a recyclable *Candida dubliniensis HIS1* marker (amplified from pMH01 and pMH02) with 80-300bp homology to the *EFG1* up and downstream regions, essentially as described [15]. Genotypes of all transformants were verified by two PCR reactions of extracted genomic DNA. The first used a forward primer upstream of the gene being deleted and a reverse primer internal to the gene, and the second used the same upstream forward primer and a reverse primer internal to the selectable marker.

Overexpression strains were generated in the SC5314, P87, and P75010 *efg1Δ/Δ* strain backgrounds which had been made sensitive to nourseothricin by recycling the *NAT1* marker at the *his1Δ/Δ* locus simultaneously with the deletion of *EFG1*. These strains were then then used to generate overexpression mutants by homozygous replacement of the 500bp upstream of the gene of interest (*BRG1*, *WOR3*, *UME6*, *RFX2*, *LYS143*, *TYE7*, or *orf19.6888)* with a selectable *NAT1* marker and the highly expressed *TDH3* promoter (amplified from plasmid pCJN542 [37]) with 80-300bp of flanking homology to the up and downstream regions of the promoter being deleted. Transformant genotypes were verified using the three-primer method as before. All primers used in this study can be found in the S6 Table.

Homozygous *wor3Δ/Δ* strains were generated in SC5314, P87 and P75010 *his1Δ::r3NAT1r3* strains via integration of the *C.d.HIS1* marker at the *WOR3* locus. Due to the large size of the native *WOR3* promoter region, ectopic complementation was achieved by using a *TDH3* promoter-*WOR3* orf PCR construct. This was transformed into *wor3Δ/Δ* SC5314, P87, and P75010 derivative strains at the *MDR1* loci using the transient CRISPR system [35] as above. The *mdr1Δ::NAT1-TDH3_{pro}WOR3* strains were reconstituted for *WOR3* function and showed similar phenotype to WT.

## Planktonic filamentation assays

Assays were performed essentially as described by Huang et al. [15]. Wild-type and *efg1Δ/Δ* strain overnight cultures in mL YPD were grown with agitation at 30˚C. Pre-warmed 5mL aliquots of RPMI+10% FBS were then inoculated to an OD600 of 0.5 from the overnight cultures and incubated at 37˚C for 4 hours at 60 rpm. Filamentation samples were then collected via centrifugation and fixed in 4% formaldehyde in 1X PBS for 15 minutes. The samples were then washed in 1X PBS twice and stained using Calcofluor-white. Imaging of cells was done using a slit-scan confocal unit on a Zeiss Axiovert 200 microscope with a Zeiss C-Apochromat 40x/1.2 NA water immersion objective. Hyphal induction was quantified first by filament unit length and then by filament to yeast ratio. Filament unit length was measured between septations or from yeast cell to filament tip in ImageJ. At least 30 filament units were measured for each sample from three different fields of view where possible.

## Biofilm formation assays

*Silicone substrate–confocal microscopy*. Biofilm formation assays were performed essentially as described previously [11, 15]. We inoculated strains from an overnight culture in YPD to an OD600 of 0.5 in 2 mL of pre-warmed RPMI+10%FBS in an untreated 12 well plate which contained a 1.4 x 1.4cm silicone square (Bentec Medical Inc., Woodland) [15]. Plates were then incubated for 90 minutes at 37˚C with shaking at 60 rpm to allow for cells to adhere to the silicone. Silicone squares were then washed in PBS and transferred to a new 12 well plate with 2mL of pre-warmed RPMI+10% FBS. Plates were then incubated for 22.5 hours for a total of 24 hours at 37˚C at 60 rpm. Biofilms were then washed with 1X PBS and fixed in 4% formaldehyde 2.5% glutaraldehyde in 1X PBS for 1 hour, rinsed with PBS and stained with Concanavalin A, Alexafluor 594 conjugate (Life Technologies) at 25μg/mL in PBS. Staining was incubated overnight at room temperature with gentle shaking at 60rpm. Biofilms were then washed with PBS and transferred to glass scintillation vials. They were then index matched in stages by washing with 100% methanol, 50% methanol/50% methyl salicylate, and finally 100% methyl salicylate. Imaging was conducted using a Zeiss Axiovert 200 microscope with slit-scan confocal optical unit with a Zeiss 40x/0.85 NA oil immersion objective.

*96 well plate—XTT reduction*. Assays were performed essentially as described by Zarnowski et al. [38]. Strains were inoculated from plates into 5 mL liquid YPD and grown overnight with agitation at 30˚C. Cells were then diluted to $OD_{600}$ 1.67 in 1 mL prewarmed RPMI+10% FBS. 2 μL diluted cells were then transferred to flat-bottom 96 well plate containing 100μL prewarmed RPMI+10% FBS and incubated at 37˚C with 60 rpm shaking for 90 minutes. Supernatant was then removed, and wells washed twice with sterile 1X PBS to remove non-adherent cells. 100 μL fresh, prewarmed RPMI+10% FBS was added to each well and incubated at 37˚C with 60 rpm shaking for 22.5hrs. Supernatant was again removed, and wells were washed as before with 1X PBS. 100 μL 1X PBS solution containing 1mg/mL XTT, and 0.025 uL 4mM menadione (dissolved in acetone) was added to each well and incubated in the dark at 37˚C with 60rpm shaking for 1 hour. Supernatants were then transferred to a new 96 well plate and

$OD_{492}$ was measured. Each assay was performed in technical triplicates to control for variability.

*96 well plate—Fluorescence microscopy*. Assays were performed essentially as described by Do et al. [11]. Strains were cultured overnight as before in YPD. 100 μL of inducing liquid medium (RPMI, RPMI +10% FBS, or YPD + 10% FBS) was prewarmed to 37°C in a 96 well plate (Greiner 96 wells Cat# 655090). Wells were then inoculated to final OD600 of 0.05 and incubated at 37°C for 90 minutes [11]. Wells were then washed twice with 1X PBS to remove non-adherent cells and 100μL fresh, prewarmed inducing medium was added to each. Plates were then incubated at 37°C with 60rpm shaking for 22.5 hours. Supernatant was then removed, and biofilms were washed again as before. Biofilms were fixed by adding 100 μL 4% formaldehyde in 1X PBS and incubating at room temperature for 1 hour. Biofilms were then washed with PBS and stained overnight using 5.5mg/mL calcofluor-white in 1X PBS. The biofilms were washed once more with PBS and then clarified in 100μL thiodiethanol. Clarified biofilms were then imaged on Keyence fluorescence microscope using PlanFluor 20X 0.45/ 8.80–7.50mm Ph1 objective with 2X digital zoom. Technical triplicate biofilms were each imaged at three or more locations within the well to ensure even sampling.

*In vivo Candida vascular catheter biofilm model*. In vivo biofilm testing was performed with a rat external jugular venous catheter model [21]. An inoculum of $10^6$ cells/ml was allowed to grow on an internal jugular catheter placed in a pathogen-free female rat (16-week old, 400 g) for 48 h. At that time, catheters were removed from the animals and biofilms were disrupted by sonication and vortexing. Viable cell counts were determined by dilution plating. Three animal and culture replicates were used per strain.

*Biofilm image processing*: The following processing pipeline was used for all biofilm Z-stack images in this study, and all steps were performed in FIJI (ImageJ v1.53). Z stacks were converted to 32-bit after which background fluorescence signal was subtracted using the "Subtract Background" function with a "rolling ball radius" of between 20–50 pixels and "smoothing disabled" option. Apical projections were generated via the "Z-project" function with the "Max intensity projection" type. To create side view images, the "Re-slice" tool was used with the "avoid interpolation" setting followed by "Z-projection" as before. Finally, images were rescaled relative to the step size of the original Z-stack with Y scale = 5.9 for confocal and Y scale = 2.68 for Keyence derived images respectively. Pseudo-coloration was achieved using the "Yellow Hot" lookup table.

## RNA extraction and RNA sequencing

Strains were cultured for RNA extraction using our published method [39]. In brief, strains were cultured overnight in 5 mL liquid YPD media with agitation at 30°C. Prewarmed 125 mL flasks with 25mL RPMI+10% FBS were then inoculated to an OD600 of 0.2. Cultures were grown for 4 hours at 37°C with shaking at 225 rpm. Triplicate RPMI+10% FBS cultures were made from the same overnight culture for each WT and *efg1Δ/Δ* strain. Cells were harvested via vacuum filtration and quickly frozen at -80°C until RNA extraction. RNA extraction was done by mechanically disrupting cells using Zirconia beads (Ambion, Fisher Scientific, Waltham) and isolating RNA using 25:24:1 phenol:chloroform:isoamyl alcohol in conjunction with a Qiagen RNeasy Mini Kit (Qiagen, Venlo, Netherlands). Total RNA was cleaned by adding 2 units of TurboDNAse (Invitrogen) to 5μg total RNA and incubating 15 minutes at 37°C. RNA was then re-extracted via acid phenol extraction and further cleaned and concentrated by column purification and elution in 15μL nuclease free water. Lexogen mRNA Sense Kit v2 was used with 2μg purified RNA according to manufacturer instructions for short amplicons. An 11 cycle PCR was done to incorporate unique barcode sequences. The completed 102

libraries were pooled evenly and run in 4 lanes of Illumina sequencing (Novogene) with an average of 16 million reads per library.

Processing of raw fastq reads was done by trimming using cutadapt (v 1.9.1), with options "-m 42 –a AGATCGGAAGAGC" so that the Illumina 3'adapter sequence could be removed. Lexogen random priming sequences were removed with options "-u 10 –u -6" according to Lexogen's instructions. After trimming, reads were mapped to the C. albicans SC5314 reference genome assembly 22 annotation gff file using tophat (v 2.0.8) with options "-no-novel-juncs" and "-G". Samtools (v 0.1.18) with options "view–h–F 256" was used to select primary alignments. Gene counts were generated using "coverageBed" from bedtools (v 2.17.0) with option "-S" for stranded alignment counting. RNAseq reads were combined for pairs of alleles and differential expression was assessed using DEseq2 (v 1.22.1) in R (v 3.5.1) using alpha = 0.05.

## Hypha-associated gene cluster analysis

Normalized transcript counts from DESeq2 for each of the 17 WT clinical isolates were used to calculate median expression level for each gene. Relative $\log_2$ fold change values for each strain were then calculated relative to the 17-strain median. Genes were then screened for those whose normalized counts exceeded 200 in addition to a relative expression level of $\pm 1$ $\log_2$ fold change resulting 2125 genes. Relative expression levels for these genes in 10 strains were then clustered using the Cluster Affinity Search Technique (Multiple Experiment Viewer 4.9.0; [40]) using Pearson correlation and a threshold of 0.8. Gene clusters were then screened for those whose profile correlated with WT planktonic filamentation ability in RPMI+FBS, 37˚C.

## Supporting information

**S1 Fig. Filamentation of wild-type strains.** WT clinical isolate strains were assayed for planktonic hyphal formation in RPMI+10% serum at 37˚C. Cultures were grown for 4 hours with 60rpm rotation. Hyphal units were measured between septa or between yeast cell and hyphal tip for 50–100 cells in 3 fields of view. Clade number is designated in parentheses. One way ANOVA was performed on Clade 1 and Clade 4 isolates and showed significant difference within each clade ($p < 0.0001$ and p = 0.0066 respectively). Unpaired T-tests were performed within Clades 2 and 3. These indicated significant differences in filament unit length within the two clades ($p < 0.0001$ and p = 0.0002 respectively).
(TIF)

**S2 Fig. Biofilm formation of wild-type strains.** WT clinical isolate strains were assayed for biofilm formation ability in RPMI + 10% FBS at 37˚C for 24hrs using the "Silicone substrate–confocal microscopy" method. Biofilm depth was measured at three different points of the biofilm for two technical replicates. Clades 1 and 4 strains were each compared via one-way ANOVA with resultant p-values of $<0.0001$ and 0.0101 respectively. Clades 2 and 3 biofilms were compared via unpaired T-tests. Clade 3 showed significant variation with a p-value of 0.0056 while Clade 2 isolates did not show significant variation.
(TIF)

**S3 Fig. Filamentation of *TDH3-TF efg1Δ/Δ* strains.** Fluorescence image of SC5314, P87, and P75010 *efg1Δ/Δ TDH3-RFX2*, *efg1Δ/Δ TDH3-TYE7*, *efg1Δ/Δ TDH3-LYS143*, and *efg1Δ/Δ TDH3-orf19.6888* mutant planktonic hyphal formation in RPMI+10%FBS 37˚C for 4hrs. Samples were fixed in 4% formaldehyde in 1X PBS and stained with Calcofluor-white. Scale bar indicates 80 μm.
(TIF)

**S4 Fig. Imaging of *TDH3-WOR3* biofilms in the P87 and P75010 backgrounds.** Biofilms were imaged after 24 hrs at 37˚C in RPMI+10% FBS for WT, *efg1Δ/Δ*, and *efg1Δ/Δ TDH3-WOR3* strains derived from P87 (Panel A) or P75010 (Panel B) using the "Silicone substrate–confocal microscopy" method. Maximum intensity apical projections of the entire biofilm (Left column) and the top half of the biofilm (Middle column) were generated for each strain. Side view projections (Right column) were also generated. Datasets for *efg1Δ/Δ* mutant images were the same as those used in Fig 2. Scale bar indicates 60 μm.
(TIF)

**S1 Table. Wild-type and *efg1Δ/Δ* mutant strain RNAseq data.** RNAseq was performed for all 17 *efg1Δ/Δ* and WT strains. Log$_2$ fold change and adjusted p-values were calculated for each respective *efg1Δ/Δ*-WT pair. Efg1 principal genes were identified by screening for those with a magnitude log$_2$ fold change of >1 and an adjusted p-value of <0.05.
(XLSX)

**S2 Table. GO term enrichment for Efg1-regulated genes.** GO term enrichment for processes was performed for the Efg1 principal Efg1-activated genes. Computationally assigned GO terms were omitted.
(XLSX)

**S3 Table. Hypha-associated gene set based on variation in filamentation among clinical isolates.** Log$_2$ fold change values for each WT clinical isolate were calculated relative to the median expression level across all strains. Cluster affinity search technique was used to identify genes whose WT vs median expression levels correlated with WT planktonic filamentation ability.
(XLSX)

**S4 Table. Gene expression assays of *efg1Δ/Δ* and *TDH3-TF efg1Δ/Δ* strains by Nanostring.** Nanostring analysis was conducted for SC5314, P87, and P75010 clinical isolate WT, *efg1Δ/Δ*, *efg1Δ/Δ TDH3-BRG1*, *efg1Δ/Δ TDH3-UME6*, *efg1Δ/Δ TDH3-WOR3*, *efg1Δ/Δ TDH3-RFX2*, *efg1Δ/Δ TDH3-LYS143*, *efg1Δ/Δ TDH3-TYE7*, and *efg1Δ/Δ TDH3-orf19.6888*.
(XLSX)

**S5 Table C. *albicans* strains used in this study.**
(XLSX)

**S6 Table. Sequences of primers used in this study.**
(XLSX)

## Acknowledgments

We are grateful to Drs. Manning Huang, Carol Woolford, and Katherine Lagree for many helpful discussions at the start of this project, to all current Mitchell lab members for their continued interest and ideas, and to Drs. Xiaorong Lin, Michelle Momany, and Zachary Lewis for advice and perspective. We are indebted to Dr. Frederick Lanni for advice about imaging and many helpful discussions throughout the project.

## Author Contributions

**Conceptualization:** Max V. Cravener, Aaron P. Mitchell.

**Data curation:** Max V. Cravener, Eunsoo Do, David R. Andes, C. Joel McManus.

**Formal analysis:** Max V. Cravener, Eunsoo Do, David R. Andes, C. Joel McManus, Aaron P. Mitchell.

**Funding acquisition:** David R. Andes, Aaron P. Mitchell.

**Investigation:** Gemma May, Robert Zarnowski.

**Methodology:** Max V. Cravener.

**Project administration:** Aaron P. Mitchell.

**Resources:** David R. Andes, C. Joel McManus, Aaron P. Mitchell.

**Supervision:** David R. Andes, C. Joel McManus, Aaron P. Mitchell.

**Writing – original draft:** Max V. Cravener.

**Writing – review & editing:** Eunsoo Do, Aaron P. Mitchell.

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
