## [Decision Letter · Decision Letter 0]

3 Dec 2022

Dear Prof. Mitchell,

Thank you very much for submitting your manuscript "Reinforcement amid genetic diversity in the Candida albicans biofilm regulatory network" for consideration at PLOS Pathogens. As with all papers reviewed by the journal, your manuscript was reviewed by members of the editorial board and by several independent reviewers. The reviewers appreciated the attention to an important topic. Based on the reviews, we are likely to accept this manuscript for publication, providing that you modify the manuscript according to the review recommendations.

All three reviewers acknowledge the amount and quality of the work presented in your manuscript, although they also express some concerns  about the extent to which the study changes our understanding of the regulatory network controlling biofilm formation compared to current models established by previous work (including your own). Nevertheless, they agree that the study stands out by the inclusion of 17 different strains for the analysis, and they highlight the discovery of Wor3 as a new positive regulator of biofilm formation and its unexpected positive relationship with Efg1 in this program.

While you should consider all suggestions made by the reviewers to improve the clarity of the manuscript and highlight the main novel aspects, please pay specific attention to the following:

What is the main basis of strain-specific differences, as opposed to commonalities, and which role may Wor3, Brg1, and Ume6 play here (reviewer 1, part III2; reviewer 3, part IIa and c)?

Please explain how overexpressed *WOR3* restores biofilm formation in the absence of Efg1 without (?) forming hyphae (reviewer 1, part III.5-6).

Discuss how Wor3 may both antagonize (in the white-opaque switch) and support (in biofilm formation) Efg1 function (reviewer 2, part II).

Sincerely,

Joachim Morschhäuser

Academic Editor

PLOS Pathogens

Alex Andrianopoulos

Section Editor

PLOS Pathogens

Kasturi Haldar

Editor-in-Chief

PLOS Pathogens

orcid.org/0000-0001-5065-158X

Michael Malim

Editor-in-Chief

PLOS Pathogens

orcid.org/0000-0002-7699-2064

Reviewer Comments (if any, and for reference):

Reviewer's Responses to Questions

**Part I - Summary**

Reviewer #1: The manuscript by Cravener et al. investigates the regulation of filamentation and biofilm formation by C. albicans, and focuses on the fact that different clinical isolates show differences in these phenotypes yet there is a shared transcriptional response mediated by Efg1. The work is interesting, of high quality, and leads to the identification of WOR3 as a novel regulator of biofilm formation, in addition to its established role in regulating white-opaque switching and gastrointestinal colonization. Several comments/suggestions are listed below on potential edits to further improve the paper.

Reviewer #2: In this study the Mitchell lab investigate the network of genes involved in C. albicans biofilm formation through the analysis of a large number (17) distinct strains , primarily clinical, with the goal of establishing what elements and connections are generically critical for the process, and which are strain dependent. This builds on previous studies that assessed a smaller number of strains.

The initial experiments involve constructing and characterizing efg1 nulls in a set of 16 sequenced clinical isolates as well as the general lab standard strain Sc5314. Although there were wide differences in hyphal formation and biofilm formation in the various strains, the efg1 nulls were uniformly poor at both making hyphae and making biofilms, independent of the capacity of the initial strain. Next RNA-seq was applied to all the WT/egf1 mutant strain pairs to establish the genes that were up or down regulated and determine if there is a core set of Efg1 regulated genes. 97 genes were established as Efg1 activated, while 13 were consistently found to be Efg1 repressed. The Efg1 up-regulated genes were GO term enriched for biofilm formation, as well as for genes that had promoters found to be bound by Efg1 in other studies. Somewhat surprisingly, we are not given any general assessment of the apparently Efg1 repressed genes.

Many of the core genes represent classic hyphal genes; this is good because it connects the work to previous studies, but at the same time it challenges the need for this study if it is primarily confirming our current understanding of the circuitry. By connecting the expression profiles with hyphal development to identify correlated genes the manuscript expands the “regulatory network” to 150 genes and identifies a number of genes whose functions are not well understood, even though they have previously been noted to have enhanced expression in hyphal cells.

The next line of research branches out from the standard picture, investigating genes that are Efg1-dependent, but not Efg1 direct targets. Because they are part of the core they are likely to have universal rather than strain-specific regulation. The obvious candidates to provide Efg1-dependence without Efg1 binding would be Efg1-dependent and bound transcription factors; these were noted to be Brg1, Orf19.6888, Rfx2 and Ume6, with Lys143, Tye7 and Wor3 just missing the cut-off for universality. These TFs were tested as candidates for regulating the core Efg1 targets that were not directly bound by Efg1 by up-regulating their expression and making it independent of Efg1 by a promoter switch; this was done for 3 of the 17 strains under study. This analysis showed little effect on core gene transcription for Lys143, Tye7, Rfx2, and Orf19.6888, while Brg1 and Ume6 activation generated up-regulation of many core genes. The effected gene include both direct Efg1 targets and indirect targets, and thus their relationship to Efg1-mediated transcription regulation is complex – Ume6 and Brg1 have already been extensively investigated in terms of hyphal development and biofilms. Of more novelty is the situation with Wor3, which also serves to bypass the efg1 block for many direct and indirect Efg1 targets. Wor3 has not been linked to biofilms, but is connected to Efg1 in the circuit controlling the white/opaque switch.

Because Wor3 opposes Efg1 in the white/opaque switch, but seems to support it in biofilm formation, the role of Wor3 is looked at more closely. Here only 2 strains are examined for the effect of a wor3 deletion on biofilm formation; biofilm formation was blocked in both in assays on RPMI+FBS media, and catheter biofilm formation was reduced in vivo. Finally, the manuscript assesses whether Wor3 involvement in biofilm formation involves Brg1; activation of Wor1 can stimulate biofilms even in a brg1 mutant.

This paper has much to offer. It provides an impressive collection of gene expression profiles influenced by Efg1 across many strains, providing a clear picture of the consistencies and variations in that regulation circuit. It identifies a new relationship between Efg1 and Wor3 in the formation of hyphae and biofilms, and it sells the point that circuitry investigated only in the “model” strain can miss important layers of regulation, and provides support for the idea that assessing 3 to 4 distinct strain backgrounds will capture the bulk of the variation and allow distinction between generic and strain specific processes.

Reviewer #3: A complex regulatory network regulating biofilm formation, itself a complex developmental event, has been defined in the predominant laboratory strain of Candida albicans, SC5314. Work from the Mitchell lab and others has suggested that reliance on a single strain to understand such networks may be unwise and here they expand the analysis of a core transcriptional regulator, Efg1, to 12 additional clinical isolates, up from five in earlier publications. They find a fair bit of noise in the Efg1 regulons (or perhaps not noise, but interstrain variations whose significance is unclear), but also a core set of ~110 genes, some of which are likely indirect. Also in common is that deletion of EFG1 greatly reduces hyphal growth and biofilm formation in all 17 strains.

Of the core genes several are transcription factors, and three seem to have significant interactions with Efg1. Brg1 and Ume6 have previously been implicated in both interactions with Efg1 and biofilm formation; Wor3 had not. All three genes significantly rescued biofilm formation in an efg1∆ mutant when overexpressed.

There are important lessons in this paper about reliance on a single strain to define expectations, which the Candida field has done for far too long. That said, this is not a new revelation and so this feels slightly incremental, especially with another paper from the same group that asked similar questions in five strains. They demonstrate here that going from five to 17 strains perhaps does not change the story that much (Fig. 10). To be fair, they had to do the experiment with all 17 strains to show they probably didn’t need to do 17 strains.

However, this represents a fair bit of work, reinforces that there are important differences between strains, identifies a new role for Wor3 (and one contrary to expectations), and expands our understanding to phenotypic and regulatory differences between strains. The subject matter is appropriate for PLoS Pathogens. There are a few concerns, however.

**Part II – Major Issues: Key Experiments Required for Acceptance**

Reviewer #1: I did not see major experiments required, except perhaps for more details on the physical properties of cells and biofilms under conditions where target genes are overexpressed under the TDH3 promoter (see below).

Reviewer #2: In the end the most solidly supported take-home message is that Wor3, like Wor1, seems to have a positive relationship with Efg1 in biofilm formation, in contrast to the negative relationship in white/opaque switching. How this “switch” occurs is not established. The title suggests that we are getting a central understanding of genetic “buffering”, but this is primarily based on speculation, and is not rigorously tested in the study. No follow-up on the repressed core genes is done – are they part of a common process, and how a TF switches from activating to repressing in this context is not discussed.

Reviewer #3: a. The interstrain variation in the Efg1 regulons is large, several times larger than the core response. The cut-offs used for significant expression differences, however, are small, log(2)>1, p<0.05. In our experience there is some noise in these experiments and I question whether these cut-offs might artificially inflate the number of Efg10-regulated genes. By my count there are 775 Efg1-regulated genes in SC5314, of which 440 (57%) are between [log(2)] 1-2. I suspect the number might be similar with other strains. I’d encourage the authors to consider whether more stringent cut-offs might bring these regulons more in line with each other.

b. There is also a limited discussion of the differences between the well-defined and more complex regulatory network defined in several labs (including these authors) and the four TFs the focus on here. Is the network fundamentally different from strain to strain? How does the model in Fig. 11 fit with the network defined in various papers from the Mitchell, Johnson, and Nobile labs?

c. The assays used are restricted to RPMI+serum (and YPD+serum in one case), a very potent biofilm-inducing condition and wor3 mutants have significant phenotype. Yet the reduction in catheter biomass is modest (if significant). This highlights differences in mutant phenotypes in different conditions and without testing these mutants (or the overexpression strains) in more conditions, they should probably temper their conclusions about the broad applicability of their network model.

**Part III – Minor Issues: Editorial and Data Presentation Modifications**

Reviewer #1: 1. There is no author summary as is normal for PLos Pathogens?

2. I was confused about the big picture logic statements in some places. The authors investigate why certain clinical isolates show differences in their biofilm/filamentation phenotypes, and yet also focus on the shared Efg1 expression program across isolates. Ultimately, they identify downstream targets of Efg1 that are transcription factors (like WOR3) and show that these factors can also promote biofilm formation even in the absence of Efg1 when ectopically expressed. I realize the publication by Do et al. focuses more on what differences are responsible for strain-to-strain variation, but discussing what is the same vs. what is different would help the reader. Ultimately, does the current study also provide insight into strain-specific differences? Are WOR3 levels different between strains and contributing to phenotypic differences?

On this note, the abstract says that “Positive control of Efg1 direct target genes by other Efg1 direct target genes – BRG1, UME6, and WOR3 – may buffer core Efg1‐responsive gene expression against the impact of genetic variation in the C. albicans species.” But is this true? Or could variation in expression of BRG1, UME6 and WOR3 be contributing to phenotypic variation?

Furthermore, I would also suggest incorporating the work from the Krysan lab that describes the biofilm network as a small world network (with high fragility).

3. Line 159 - “We hypothesized that direct targets may govern expression of indirect targets”. How else would indirect targets be regulated other than via direct targets? I note that this is also the model proposed in Supplemental Figure S5 from Nobile et al, Cell, 2012, where the core network TFs activate downstream genes directly or via other TFs.

4. On this note, it is perhaps worth mentioning that WOR3 was not picked up in the Nobile screen as it was not one of the factors included in their TF library.

5. What is the structure of the TDH3-WOR3 (efg1 KO) biofilm given that this strain does not form hyphae and yet forms biofilms similar in mass to the wildtype control? Is this forming a robust biofilm (Fig. 7) without making normal hyphae? Or are hyphae forming under biofilm conditions but not in planktonic conditions (Fig. 6). This is hard to tell from the images in Fig. 8.

6. It would actually be great to have some quantification of the hyphal cells (percentage/length?) in the TDH3-driven strains.

7. There appear to be (undeclared) splices in the lateral biofilm images in figures 2, 8 and 9. These should be modified to be clearer as to what is shown.

Additional points

It is slightly surprising that TDH3-driven expression levels are not much higher than those observed from the natural promoter given that TDH3 is a very strong promoter?

Line 169. “As a readout, we used Nanostring probes to measure expression of several indirect and direct targets (S6 Table).” Would be good to be precise here and to give the number of genes being analyzed by Nanostring. Looks like it is 103 genes from the table?

Would it make sense to split Figure 4 into direct and indirect targets of Efg1? Just to highlight the genes from both groups? This data is in the table but a graphical summary would give the reader a clearer picture of the pattern of regulation.

For Figure 4, I believe that all of the expression data is from strong hyphal-inducing conditions. This could be made stated in the text and in the legend.

Figure 5 could be moved to supplemental material.

Line 322 “However, a Efg1 core17‐exp direct target like HWP1 will be Efg1‐responsive even if Efg1 itself cannot activate HWP1 by direct binding because of the intervention of Brg1, Wor3, and Ume6.” Suggest adding the underlined words to make clear what is meant.

Throughout the text and figures (e.g. Figure 11) it could be emphasized that direct targets of Efg1 refer to direct binding of Efg1 to those promoters, as this may not always be obvious to all readers.

Reviewer #2: (No Response)

Reviewer #3: Fig. 1 could benefit from either headings (WT/efg1∆/∆) or a panel A/B. Or an organization more like Fig. 2.

PLOS authors have the option to publish the peer review history of their article (what does this mean?). If published, this will include your full peer review and any attached files.

Reviewer #1: No

Reviewer #2: No

Reviewer #3: No

Figure Files:

Data Requirements:

Reproducibility:

References:

---

## [Editor Report · Decision Letter 1]

9 Jan 2023

Dear Professor Mitchell,

We are pleased to inform you that your manuscript 'Reinforcement amid genetic diversity in the Candida albicans biofilm regulatory network' has been provisionally accepted for publication in PLOS Pathogens.

Best regards,

Joachim Morschhäuser

Academic Editor

PLOS Pathogens

Alex Andrianopoulos

Section Editor

PLOS Pathogens

Kasturi Haldar

Editor-in-Chief

PLOS Pathogens

orcid.org/0000-0001-5065-158X

Michael Malim

Editor-in-Chief

PLOS Pathogens

orcid.org/0000-0002-7699-2064
---

## [Editor Report · Acceptance letter]

18 Jan 2023

Dear Professor Mitchell,

We are delighted to inform you that your manuscript, "Reinforcement amid genetic diversity in the Candida albicans biofilm regulatory network," has been formally accepted for publication in PLOS Pathogens.

Best regards,

Kasturi Haldar

Editor-in-Chief

PLOS Pathogens

orcid.org/0000-0001-5065-158X

Michael Malim

Editor-in-Chief

PLOS Pathogens

orcid.org/0000-0002-7699-2064